

# Improving Seasonally Frozen Ground Monitoring Using Soil Freezing Characteristic Curve in Permittivity-Temperature Space

Hesam Salmabadi[1,2], Renato Pardo Lara[3], Aaron Berg[3], Alex Mavrovic[1,2,4], Chelene Hanes[5], Benoit Montpetit[6], and Alexandre Roy[1,2]

[1]Department of Environmental Sciences, University of Quebec in Trois-Rivières, Trois-Rivières, Quebec, Canada
[2]Centre d'Études Nordiques, Université Laval, Québec, Quebec, Canada
[3]Department of Geography, Environment & Geomatics, University of Guelph, Guelph, Ontario, Canada
[4]Department of Physics, Cégep de Sherbrooke, Sherbrooke, Quebec, Canada
[5]Great Lakes Forestry Centre, Canadian Forest Service, Natural Resources Canada, Sault Ste. Marie, Ontario, Canada
[6]Climate Research Division, Environment and Climate Change Canada, Toronto, Ontario, Canada

**Correspondence:** Hesam Salmabadi (hesam.salmabadi@uqtr.ca)

**Abstract.** Frozen ground, a key indicator of climate change, profoundly influences ecological, hydrological, and carbon flux processes in cold regions. However, traditional monitoring methods, which rely on a binary $0°C$ soil temperature threshold, fail to capture the complexities of soil freezing, such as freezing point depression and transitional states where water and ice coexist. This study introduces a framework that fits a theoretical Soil Freezing Characteristic Curve (SFCC) in permittiv-

ity–temperature space to site- and cycle-specific in situ measurements. This approach enables the quantification of the degree of soil freezing and the classification of soil states as frozen, unfrozen, or in transition (partially frozen). We analyzed 135 freezing cycles from 87 sites, each equipped with permittivity-based soil moisture probes. These sites are part of eight monitoring networks spanning diverse Canadian landscapes, including eastern boreal forests (Montmorency Forest, La Romaine, James Bay, Chapleau), western boreal forests (Candle Lake), prairies (Kenaston), and tundra regions (Trail Valley Creek and

George River). On average, eastern boreal forest sites exhibited prolonged unfrozen and transitional states due to high soil moisture retention and insulation from snow and vegetation cover (23 frozen days, 46 transitional days). In contrast, western boreal forest sites experienced more extensive freezing under drier conditions (73 frozen days, 76 transitional days). Prairie sites displayed equal durations of frozen and transitional states (71 days each), while tundra sites had the longest frozen periods (145 frozen days, 52 transitional days). Notably, transitional periods lasted as long as—or even longer than—frozen ones,

underscoring the limitations of binary classifications. Furthermore, the traditional $0°C$ threshold misclassified transitional soil states, overestimating frozen days by over 87% in prairie and western boreal regions, and unfrozen days by 86% in the eastern boreal forest. In tundra, the bias was more balanced, with 64% and 36% of transitional days misclassified as unfrozen and frozen, respectively. This SFCC-based framework enhances seasonally frozen ground monitoring, offering deeper insights into soil freeze-thaw dynamics. These advancements have implications for improving climate change assessments, refining car-

bon flux models, and training and validating remote sensing products. Additionally, the resulting database of soil states from this study provides a valuable resource for advancing frozen ground research, particularly in remote sensing and ecosystem modeling efforts.



# 1 Introduction

Frozen ground is defined as a condition in which pore water (the water inside the soil) turns into ice (Williams and Smith,
1989). Recognized as a key climate change indicator by the Environmental Protection Agency (EPA), this phenomenon is
widespread and affects most land areas above $45°$N latitude (Zhang et al., 2003). Frozen ground plays a major role in land
surface energy and water balance, thereby impacting all ecological, hydrological, pedological, and biological activities in these
regions (Ala-Aho et al., 2021; Hayashi, 2013; Ping et al., 2015; Loranty et al., 2018). By affecting hydrological partitioning,
it regulates key processes such as infiltration, percolation, groundwater recharge, water chemistry, runoff characteristics, and
evapotranspiration (Ala-Aho et al., 2021). Frozen ground impacts soil respiration, the primary pathway of carbon emissions
to the atmosphere, as it is largely regulated by soil temperature and water content (Davidson and Janssens, 2006; Lei et al.,
2022; Nikrad et al., 2016; Arndt et al., 2023; Azizi-Rad et al., 2022; Mikan et al., 2002; Mavrovic et al., 2023). Therefore,
understanding frozen ground—its timing, extent, and duration—is critical for tracking climate change impacts and predicting
changes in these vital environmental processes.

Traditionally, the state of the soil has been defined through a single measurement of soil temperature within the top few
centimeters of the ground (Andersland and Ladanyi, 2003). Soil is labeled as frozen if the soil (or air temperature) is below $0°$C
and unfrozen if above. This approach is widely used in numerous studies, particularly in remote sensing, to monitor seasonally
frozen ground conditions (Kim et al., 2011; Taghipourjavi et al., 2024; Zhang and Armstrong, 2001; Gao et al., 2020; Kou et al.,
2017). However, the soil freezing and thawing process is not binary, and using a single threshold of $0°$C—the freezing point of
pure water—is not sufficiently accurate (Pardo Lara et al., 2020; Mavrovic et al., 2020). In natural environments, soil typically
begins to freeze at temperatures below $0°$C (Dobiński, 2020), a phenomenon known as soil freezing point depression, which
occurs due to adsorption, capillary action, adhesive and cohesive forces, and osmotic effects (Tian et al., 2014; Bouyoucos
and McCool, 1915). Moreover, freezing occurs over a range of temperatures due to the presence of various types of water in
the soil—hygroscopic (unfreezable) water, capillary water, water in soil pores, and gravitational water—all of which behave
differently during freezing. This diversity creates a transitional zone where water and ice coexist, challenging the traditional
binary classification (Tian et al., 2014; Zhang et al., 2019; Bouyoucos and McCool, 1915). Considering soil freezing point
depression, rather than $0°$C threshold, would enhance the accuracy of detecting the soil's state, thereby minimizing false
positives and negatives in assessments. Additionally, detecting the transitional period is crucial for accurately determining
"zero curtain" periods. The zero curtain, observed during shoulder seasons, is a phase when soil temperatures hover around
the soil's freezing point regardless of air temperatures. This period is critical as the soil's near-freezing temperature sustains
microbial activity (Schimel and Mikan, 2005), significantly impacting carbon dioxide fluxes (Arndt et al., 2023). Studies show
that carbon emissions during the zero curtain in the fall can match or exceed those of the rest of winter (Arndt et al., 2023;
Mavrovic et al., 2023). Furthermore, an observed increase in the duration of the zero curtain period extending into the fall and
winter seasons leads to higher carbon emissions during the non-growing season, underscoring the importance of accurately
identifying these periods (Arndt et al., 2023).





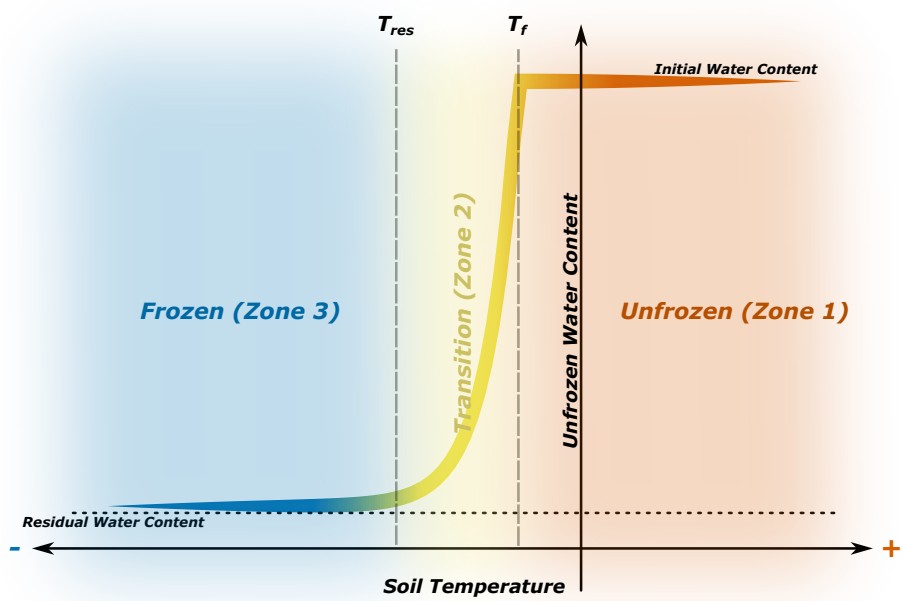

**Figure 1.** A typical Soil Freezing/Thawing Characteristic Curve (SFCC/STCC), adapted from Zhang et al. (2019). $T_f$ marks the temperature at which soil begins to freeze, while $T_{res}$ indicates the temperature at which the liquid water content stabilizes, nearly equal to the residual water content.

The Soil Freezing/Thawing Characteristic Curve (SFCC/STCC), which defines the relationship between liquid water content ($\theta_{\text{lw}}$) and subzero temperatures (Spaans and Baker, 1996; Koopmans and Miller, 1966), offers a framework for capturing the complexities of the soil freezing and thawing process (Pardo Lara et al., 2020). A typical SFCC/STCC (Fig. 1) consists of three zones (Zhang et al., 2019). Zone 1, called the "unfrozen zone," occurs when the soil temperature is above the freezing point. In this zone, the pore water content remains almost constant, regardless of temperature changes. Zone 2, which starts immediately after the freezing point ($T_f$), is the "transitional zone." Here, there is a sharp decrease in pore water content as the soil-water system reaches a balance between liquid water, and ice. Zone 3 begins at the soil temperature where the liquid water content stabilizes, referred to as $T_{\text{res}}$ (Kozlowski, 2007; Chen et al., 2021). In this zone, both bulk and capillary water content drop to zero, and the remaining water exists as a thin film bound to soil particles, commonly called unfreezable or bound water (Chen et al., 2021). To construct the SFCC, measurements of liquid water content and soil temperature are required. While measuring soil temperature is relatively straightforward, accurately quantifying the liquid water content in frozen soil poses significant challenges, especially under field conditions.

Dielectric-based methods, such as Time Domain Reflectometry, Capacitance, and Impedance techniques, are widely used to estimate $\theta_{\text{lw}}$ due to their ability to provide continuous, non-destructive in situ measurements (Seyfried and Murdock, 1996; Michael W Smith and Tice, 1988). These methods exploit the high permittivity contrast between liquid water ($\varepsilon_{\text{lw}} \approx 80$) and other soil components—air ($\varepsilon_{\text{air}} \approx 1$) and soil minerals ($\varepsilon_{\text{soil}} \approx 5$)—allowing for efficient estimation of $\theta_{\text{lw}}$. Typically,



physically based models, such as dielectric mixing models, or empirical models are employed to relate soil effective (bulk) permittivity ($\varepsilon_{\text{eff}}$), measured by dielectric probes—which represents the combined dielectric response of soil components—to $\theta_{\text{lw}}$. However, relating $\varepsilon_{\text{eff}}$ to $\theta_{\text{lw}}$ in frozen soils remains challenging. Dielectric mixing models require accurate ice content

estimates, which are difficult to obtain in situ, often leading to overestimations of $\theta_{\text{lw}}$ when the ice component is neglected (Amankwah et al., 2022; Zhou et al., 2014; He and Dyck, 2013). This overestimation occurs because frozen soils, at identical liquid water contents, exhibit a higher $\varepsilon_{\text{eff}}$ than unfrozen soils due to the higher permittivity of ice ($\varepsilon_{r_{\text{ice}}} \approx 3.2$) compared to air ($\varepsilon_{r_{\text{air}}} \approx 1$) (Spaans and Baker, 1995). Similarly, empirical models often rely on calibrations developed for unfrozen soils, limiting their applicability to frozen conditions (Yoshikawa and Overduin, 2005; He and Dyck, 2013). Calibration equations

specifically developed for frozen soils are often restricted to saturated soils (Michael W Smith and Tice, 1988) or specific soil types, requiring prior knowledge of the soil's total water content before freezing (Spaans and Baker, 1995). These limitations make constructing the SFCC problematic, particularly in dynamic field environments. To address this, we constructed the SFCC directly in permittivity–temperature space, bypassing the need to estimate $\theta_{\text{lw}}$. This approach improves the accuracy of freeze–thaw characterization while overcoming the shortcomings of previous studies, which used logistic curve fitting with

limited physical interpretability and robustness (Pardo Lara et al., 2020).

This study aims to enhance seasonally frozen ground monitoring by utilizing an SFCC model in the permittivity–temperature space. By fitting this model to in situ measurements across diverse Canadian ecosystems—including prairies, boreal forests, and tundra regions—and constructing site- and cycle-specific SFCCs, we examine freezing and thawing processes in soils and define soil states under varying properties and climatic conditions. This work lays the foundation for improving remote sensing

algorithms and ecosystem models, offering a more nuanced representation of soil state transitions beyond traditional binary approaches.

## 2   Materials and Methods

### 2.1   Overview of Methodology

Before detailing the materials and methods, we provide an overview of the process. First, in situ data were collected (Sect. 2.2)

and preprocessed (Sect. 2.3) to identify individual freezing and thawing cycles at each site. The SFCC model was then fitted (Sect. 2.4) to in situ soil temperature and $\varepsilon_{\text{eff}}$ data using curve fitting, allowing for the determination of model parameters and the construction of site- and cycle-specific SFCCs. Finally, a Monte Carlo approach was applied to propagate measurement and parameter uncertainties, enabling the quantification of soil freezing and the classification of soil states for each site and cycle (Sect. 2.5). This framework facilitated continuous monitoring of soil freeze–thaw dynamics.

### 2.2   Data Collection

This study leverages in situ measurements of $\varepsilon_{\text{eff}}$ and soil temperature to investigate the soil freezing and thawing processes and monitor the soil state. Our research spans eight networks located in diverse environments (Table 1 and Fig. 2), including





eastern boreal forests (Montmorency Forest, La Romaine, James Bay, Chapleau), western boreal forest (Candle Lake), prairies (Kenaston), and tundra (Trail Valley Creek, George River). Three types of soil moisture sensors—HydraProbe (Stevens Water), TEROS-12 (METER Group), and CS616 (Campbell Scientific)—were deployed based on availability across the monitoring networks. HydraProbe and TEROS-12 simultaneously measure soil temperature and moisture, whereas the CS616 requires a separate soil temperature sensor (CS109SS-L) for temperature measurements. We provided an overview of these sensors, including their measurement principles and operational specifics, in Appendix B. At most sites, soil moisture probes were horizontally inserted to a depth of 5 cm, primarily within the organic layer unless it was notably thin, in which case measurements extended into the mineral layer. For the TEROS-12 and HydraProbe, insertion was horizontal at this depth. In the Chapleau network, the CS616, with its 30 cm needle, was inserted at a 20-degree angle to integrate measurements over the top 10 cm of soil, with the midpoint depth set at 5 cm, thereby ensuring comparability with other sites (Fig. A1). Each site typically features a single sensor, except for Chapleau and Trail Valley Creek. Chapleau includes 24 CS616 sensors distributed across four 200 × 200 m plots, each representing a different forest type (Hanes et al., 2023). Since only one soil temperature sensor was available in the middle of each plot, data from multiple CS616 sensors were averaged to represent the soil conditions of that plot. At Trail Valley Creek, 12 HydraProbes were distributed across six sites (Montpetit et al., 2024, 2025), and data from each sensor were analyzed separately. Site- and sensor-specific calibration was not performed prior to installation; however, the use of manufacturer specifications and previously published validation studies provides confidence in the measurement accuracy of the deployed sensors within their expected uncertainty ranges (Seyfried and Murdock, 2004; Pardo Lara et al., 2021; Kelleners et al., 2005; Logsdon, 2009; Cominelli et al., 2024; Fragkos et al., 2024). A general quality control procedure was applied by removing physically implausible values, defined as soil temperatures outside the range of $-60$ to $60\,°C$ and permittivity values outside the range of 1 to 90. All in situ measurements were then subjected to a standardized preprocessing workflow. Data were resampled to a uniform hourly resolution, and a continuous time index was constructed to ensure temporal consistency. Missing values were filled using linear interpolation; however, interpolated values were used solely for categorical soil state labeling and were excluded from all curve-fitting analyses to preserve the integrity of model-derived parameters. The distribution of sites within each network is strategically designed to capture the ecological diversity and maximize spatial variability, influenced by the challenging terrain and limited accessibility of the network areas. Our goal is to comprehensively represent the diversity of each network, thereby enhancing our understanding of regional soil dynamics. Detailed characteristics for each site—including geographic coordinates, soil textural composition (clay, sand, and silt percentages), organic content, specific soil types, elevation and stratification of soil layers—are are available through an interactive map hosted in a GitHub repository (Soil-Temperature-Permittivity-Monitoring-Sites) (Salmabadi et al., 2025).

## 2.3 Data Preprocessing

The data preprocessing stage began by converting the raw outputs of the sensors into $\varepsilon_{\mathrm{eff}}$ for the bulk soil. This process varied depending on the sensor type, as each sensor produced different raw outputs. Detailed explanations of these sensor-specific preprocessing steps are provided in Appendix B. Next, we identified freezing and thawing cycles based on trends in soil temperature and $\varepsilon_{\mathrm{eff}}$. A freezing cycle was defined as the period when soil temperature started decreasing and reached its



**Table 1.** Summary of soil moisture sensor deployment across our networks

| Land Cover | Network Name | Sensor Type[†] | Sensor Depth (cm) | Number of Sites (Probes) | Temporal Coverage |
|---|---|---|---|---|---|
| Eastern boreal forest | James Bay | TEROS 12 | 5 | 9 (9) | 2020-08 to 2022-06 |
| | Montmorency Forest | TEROS 12 | 5 | 10 (10) | 2020-10 to 2024-06 |
| | La Romaine | TEROS 12 | 5 | 3 (3) | 2022-08 to 2023-07 |
| | Chapleau | CS616 | 0–10[†] | 4 (24) | 2017-08 to 2022-06 |
| Western boreal forest | Candle Lake | HydraProbe | 5 | 17 (17) | 2022-08 to 2023-05 |
| Prairies | Kenaston | HydraProbe | 5 | 37 (37) | 2014-08 to 2018-06 |
| Tundra | Trail Valley Creek | HydraProbe | 5 | 6 (12) | 2018-09 to 2019-03 |
| | George River | TEROS 12 | 5 | 1 (1) | 2022-09 to 2023-07 |

† All sensors include built-in soil temperature measurement, except the CS616, which requires a separate soil temperature sensor (CS109SS-L).

† The CS616 is angled at 20°, measuring the top 10 cm of soil with a midpoint at 5 cm.

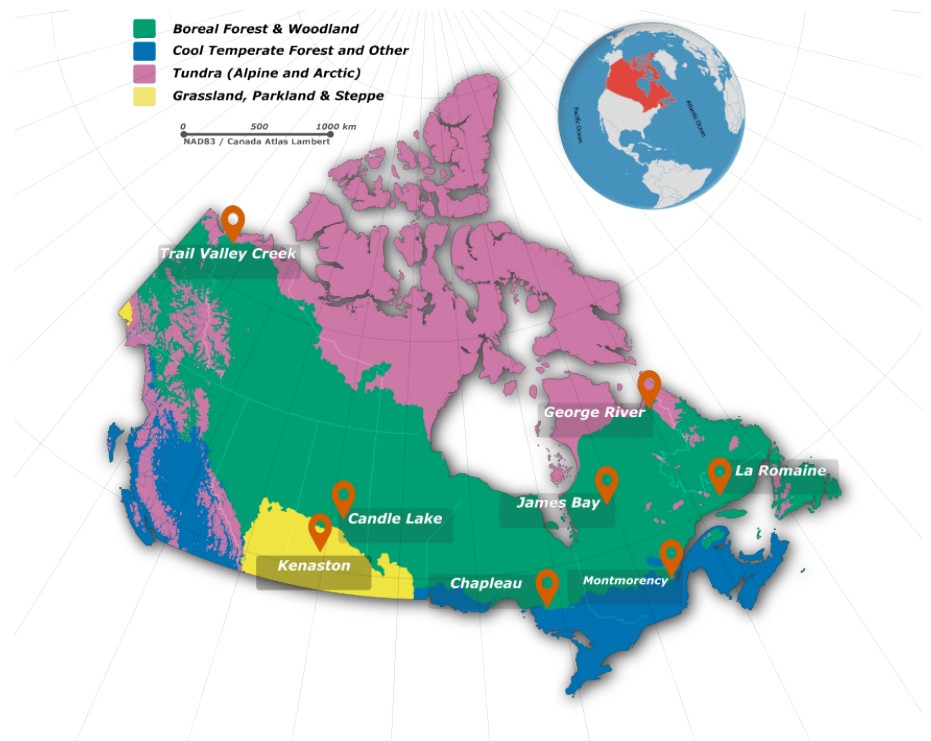

**Figure 2.** Geographic distribution of monitoring networks across Canadian landscapes. The land cover map is adapted from Latifovic (2019).

minimum, while a thawing cycle extended from this minimum until temperatures rose above zero. This approach aimed to capture the main freezing and thawing events while excluding minor fluctuations, where temperatures briefly rose above or fell below zero within a narrow range, resulting in incomplete or transient freezing or thawing. Specifically, any fluctuations



within $\pm 2\sigma_T$ of $0°$C, where $\sigma_T$ represents the instrument-specific temperature uncertainty, were ignored (see Appendix B for details on sensor uncertainty). If, during a freezing cycle, the soil temperature never dropped below the $-\sigma_T$ threshold and $\varepsilon_{\text{eff}}$ remained relatively unchanged, we classified these sites as never frozen. Although curve fitting was not feasible for these sites due to insufficient data in Zones 2 and 3, they were retained for further analysis to investigate the freezing process across our monitoring networks. In this analysis, we assumed that the total water content in the system remained equal to

the initial water content and did not change during the freezing or thawing processes (He and Dyck, 2013). We monitored $\varepsilon_{\text{eff}}$ throughout both freezing and thawing cycles to validate this assumption. We interpreted significant, sudden surges in $\varepsilon_{\text{eff}}$ as indicators of additional water entering the system, violating this assumption. Consequently, we excluded such cycles from further analysis. While this assumption generally held during freezing cycles, it was often invalid during thawing cycles, primarily due to snowmelt introducing substantial amounts of water into the soil. As a result, the SFCC could be reliably

constructed for freezing cycles, but constructing the STCC from in situ measurements during thawing cycles was often not feasible. Since this study aimed to use site- and cycle-specific SFCC/STCC to determine the degree of soil freezing and classify soil states, we applied the SFCC from each freezing cycle to the subsequent thawing cycle at the same site. Although differences between SFCC and STCC—primarily driven by hysteresis effects—are pronounced in laboratory settings, previous studies (Pardo Lara et al., 2020; Mavrovic et al., 2020) found no visually discernible differences in situ. We assumed that the

SFCC could reliably represent the STCC until a significant change in total soil water content, marked by a sudden surge in $\varepsilon_{\text{eff}}$, which approximately corresponds to the snowmelt event. To assess the reliability of applying the SFCC for thawing cycles, we evaluated its performance over the full thawing cycle and up to the snowmelt date. The final step in the preprocessing of in situ data involved ensuring a balanced dataset to prevent overfitting during the curve-fitting process. In practice, the distribution of data across temperature ranges was often uneven, which could bias the fitting process. We averaged $\varepsilon_{\text{eff}}$ values within a $0.1°$C

temperature range to address this imbalance, corresponding to the sensors' temperature resolution. This approach not only compensated for the uneven distribution but also reduced noise caused by diurnal temperature fluctuations. Such fluctuations, while absent in controlled laboratory settings, were common in in situ environments.

## 2.4    Data Processing

### 2.4.1    SFCC Model

In this study, we applied the theoretical model introduced by Bai et al. (2018) to construct the SFCC, which estimates liquid water content as a function of soil temperature:

$$
\theta_{lw} = \begin{cases} (\theta_{\text{int}} - \theta_{\text{res}})e^{b(T_{\text{soil}} - T_f)} + \theta_{\text{res}} & \text{if } T_{\text{soil}} < T_f \\ \theta_{\text{int}} & \text{if } T_{\text{soil}} \geq T_f \end{cases} \tag{1}
$$





where $\theta_{lw}$, $\theta_{\text{int}}$, and $\theta_{\text{res}}$ represent the liquid, total (initial water content prior to freezing), and residual water content, respectively. $T_{\text{soil}}$ is the soil temperature, $T_f$ is the freezing point, and $b$ controls the steepness of the transition curve, representing the shape factor of the distribution function ($^{\circ}\text{C}^{-1}$).

The relationship between the $\varepsilon_{\text{eff}}$ and $\theta_{lw}$ can be physically derived using mixing models (Amankwah et al., 2022; Kelleners and Norton, 2012; Roth et al., 1990), which express effective (bulk) permittivity as a volume-weighted average of soil components:

$$\varepsilon_{\text{eff}}^{\alpha} = \theta_{lw}\varepsilon_{lw}^{\alpha} + \theta_{\text{ice}}\varepsilon_{\text{ice}}^{\alpha} + (n - \theta_{lw} - \theta_{\text{ice}})\varepsilon_{\text{air}}^{\alpha} + (1-n)\varepsilon_{\text{soil}}^{\alpha} \tag{2}$$

where $n$ represents the soil porosity, and $\varepsilon_{\text{eff}}$, $\varepsilon_{\text{soil}}$, $\varepsilon_{lw}$, $\varepsilon_{\text{ice}}$, and $\varepsilon_{\text{air}}$ are the relative dielectric permittivities (dimensionless) of bulk soil, soil solids, liquid water, ice, and air, respectively. The parameter $\alpha$ depends on soil structure and composition, ranging from -1 (parallel arrangement) to 1 (series arrangement), with $\alpha \neq 0$ (Amankwah et al., 2022). Solving for $\theta_{lw}$ gives:

$$\theta_{lw} = \frac{\varepsilon_{\text{eff}}^{\alpha} - (1-n)\varepsilon_{\text{soil}}^{\alpha} - n\varepsilon_{\text{air}}^{\alpha} - \theta_{\text{ice}}\left(\varepsilon_{\text{ice}}^{\alpha} - \varepsilon_{\text{air}}^{\alpha}\right)}{\varepsilon_{lw}^{\alpha} - \varepsilon_{\text{air}}^{\alpha}} \tag{3}$$

This equation defines the relationship between $\theta_{lw}$ and $\varepsilon_{\text{eff}}$, where $\varepsilon_{\text{air}}$, $\varepsilon_{lw}$, and $\varepsilon_{\text{ice}}$ are known constants with minimal temperature dependence, while $n$, $\alpha$, and $\varepsilon_{\text{soil}}$ vary with soil composition and structure and are treated as unknowns. We modified the model by Bai et al. (2018) (Eq. 1) by incorporating Eq. (3) to implement the SFCC in permittivity-temperature space. The detailed solution process is presented in Appendix B. The modified equation is:

$$\varepsilon_{\text{eff}}(T) = \begin{cases} \left((\varepsilon_{\text{int}}^{\alpha} - \varepsilon_{\text{res}}^{\alpha})e^{b(T_{\text{soil}}-T_f)} + \varepsilon_{\text{res}}^{\alpha}\right)^{\frac{1}{\alpha}} & \text{if } T < T_f \\ \varepsilon_{\text{int}} & \text{if } T \geq T_f \end{cases} \tag{4}$$

Here, $\varepsilon_{\text{int}}$ represents the pre-freezing $\varepsilon_{\text{eff}}$, corresponding to the system's total water content, which is assumed to approximate the initial water content (He and Dyck, 2013). $\varepsilon_{\text{res}}$ is the $\varepsilon_{\text{eff}}$ associated with the residual water content. The parameter $\alpha$, as mentioned earlier, is an exponent that depends on the soil structure and composition. To fully define the SFCC, it is necessary to estimate $T_{\text{res}}$, which marks the onset of Zone 3 in the SFCC (Fig.1). From Eq. (4), we have:

$$\varepsilon_{\text{eff}}(T)^{\alpha} = (\varepsilon_{\text{int}}^{\alpha} - \varepsilon_{\text{res}}^{\alpha})e^{b(T-T_f)} + \varepsilon_{\text{res}}^{\alpha} \tag{5}$$

As $T \to -\infty$, $\varepsilon_{\text{eff}}(T)$ approaches $\varepsilon_{\text{res}}$. To determine the temperature at which $\varepsilon_{\text{eff}}(T)$ approaches $\varepsilon_{\text{res}}$, we introduce a small threshold $\delta$:

$$(\varepsilon_{\text{int}}^{\alpha} - \varepsilon_{\text{res}}^{\alpha})e^{b(T-T_f)} < \delta \tag{6}$$

Solving for $T$:



$$T < \frac{\ln\left(\frac{\delta}{\varepsilon_{\text{int}}^{\alpha} - \varepsilon_{\text{res}}^{\alpha}}\right)}{b} + T_f \tag{7}$$

Thus, the temperature $T_{\text{res}}$, where $\varepsilon_{\text{eff}}(T)$ reaches the plateau, is approximately:

$$T_{\text{res}} \approx \frac{\ln\left(\frac{\delta}{\varepsilon_{\text{int}}^{\alpha} - \varepsilon_{\text{res}}^{\alpha}}\right)}{b} + T_f \tag{8}$$

In our calculations, we used $\delta = 0.01$, which allows us to determine the temperature where the difference between $\varepsilon_{\text{eff}}(T)$ and $\varepsilon_{\text{res}}$ is negligible.

In summary, by incorporating a dielectric mixing model into the theoretical framework proposed by Bai et al. (2018) (Eq. 1), we constructed the SFCC in the permittivity–temperature space. The resulting SFCC model (Eq. 4) calculates $\varepsilon_{\text{eff}}$ as a func-

tion of soil temperature, using key parameters such as the shape factor of the distribution function $b$ (1/°C), freezing point depression ($T_f$), and the initial ($\varepsilon_{\text{int}}$) and residual ($\varepsilon_{\text{res}}$) permittivity, which are determined using optimization and curve-fitting techniques. Finally, calculating $T_{\text{res}}$ (Eq. 8) identifies the temperature at which the soil is completely frozen, defining the onset of Zone 3 in the SFCC and completing its characterization.

### 2.4.2 Parameter Estimation for the SFCC Model

To derive the model parameters—$\varepsilon_{\text{int}}, \varepsilon_{\text{res}}, b,$ and $T_f$—and construct the SFCC, we applied a systematic data processing and curve-fitting approach tailored to our SFCC model (Eq. 4). To ensure accuracy and physical plausibility, we used non-linear least squares optimization with initial guesses and parameter bounds. The curve fitting was conducted using the *curve_fit* function from the SciPy library (version 1.13.1) in Python (Virtanen et al., 2020), employing the Trust Region Reflective (TRF) algorithm, which optimizes parameter values while keeping them within predefined bounds. We fitted the SFCC model

(Eq. 4) to each individual freezing cycle, generating site- and cycle-specific SFCCs to estimate the degree of soil freezing. Since the fitting is performed independently for each site and cycle, variations in absolute $\varepsilon_{\text{eff}}$ values, caused by differences in probe operating frequencies—due to the frequency-dependent dielectric properties of water and bulk soil—do not affect soil state monitoring.

To ensure that the data used for model fitting primarily reflect the freezing process, we included only measurements where

$T_{\text{soil}} \leq 2$°C, capturing temperatures where freezing processes are actively occurring. During initial analyses, $\alpha$, the exponent representing soil structure in the dielectric mixing model (Eq. 3), consistently converged to boundary values without improving model fit, indicating low sensitivity. This issue likely stems from eliminating parameter $B$ when deriving the SFCC model from Bai et al. (2018)'s framework (Eq. B17). This reduction decreases the model's dependence on soil parameters such as porosity and soil solid permittivity, both of which influence $\alpha$. To enhance model stability and interpretability, we fixed $\alpha$ at

0.5, a commonly used value (Pardo Lara et al., 2020; Seyfried et al., 2005). We set the lower bound of $\varepsilon_{\text{res}}$ at 1, the lowest measurable probe range, and constrained the upper bound to remain below $\varepsilon_{\text{int}}$ to prevent unrealistic values. The initial effective



permittivity, $\varepsilon_{\text{int}}$, representing pre-freezing soil permittivity, was initialized as the mean $\varepsilon_{\text{eff}}$ within $\sigma_T°\text{C} \leq T_{\text{soil}} \leq 2°\text{C}$, where $\sigma_T$ represents the instrument-specific temperature uncertainty (see Appendix B for details on sensor uncertainty). This range ensures that mainly unfrozen-state data (Zone 1) contribute to the estimate, accounting for sensor uncertainty. The bounds

for $\varepsilon_{\text{int}}$ were set as the observed minimum and maximum $\varepsilon_{\text{eff}}$ within this range. The exponential constant $b$, which controls the transition from $\varepsilon_{\text{int}}$ to $\varepsilon_{\text{res}}$, was initialized at 1.0, with a lower bound of 0.1 to prevent overly gradual transitions and no upper bound. For the freezing temperature $T_f$, we allowed values up to $+1°\text{C}$ to accommodate known measurement biases and sensor discrepancies. For instance, Pardo Lara et al. (2020) suggested that dielectric sensors may detect permittivity changes indicative of freezing before thermistors register subzero temperatures, likely due to differences in placement, thermal inertia,

or measurement volume (Pardo Lara et al., 2021). These discrepancies can cause $T_f$ to register as slightly positive, particularly during rapid temperature transitions.

To assess the robustness and uncertainty of our fitted parameters ($\varepsilon_{\text{int}}, \varepsilon_{\text{res}}, b, T_f$), we applied bootstrapping, resampling in situ soil temperature and $\varepsilon_{\text{eff}}$ data 1,000 times. To ensure representative selection across temperature ranges, we divided the data into blocks and resampled within each block with replacement. This approach preserved the natural distribution of $\varepsilon_{\text{eff}}$

across the soil temperature range while introducing variability across iterations. Each iteration generated a new dataset for curve fitting, enabling us to quantify parameter variability, construct confidence intervals, and identify potential biases. The resulting bootstrapped parameter distributions were used to compute mean values and standard deviations.

## 2.5   Data Postprocessing

### 2.5.1   SFCC-based Soil State Classification

Although the SFCC could be constructed directly using the parameters from Sect. 2.4.2, accounting for measurement and model uncertainties was essential. To do so, a Monte Carlo framework with $N = 15,000$ simulations was employed for each measurement point (i.e., each hourly soil temperature observation). Soil temperature measurements were modeled as normally distributed, $T_{\text{sim}} \sim \mathcal{N}(T_{\text{obs}}, \sigma_T^2)$, with sensor-specific standard deviations. We quantified parameter uncertainties using bootstrap statistics. Each variable ($M$)—where $M \in \{T_f, T_{\text{res}}, \varepsilon_{\text{int}}, \varepsilon_{\text{res}}, b\}$—was sampled from a normal distribution, $\mathcal{N}(\mu_M, \sigma_M^2)$

with $\mu_M$ and $\sigma_M$, obtained from bootstrap analysis. We discarded simulations that violated physical constraints (i.e., $\varepsilon_{\text{int}} < \varepsilon_{\text{res}}$ or $T_{\text{res}} > T_f$) to ensure result validity. For each simulation, an effective permittivity, $\varepsilon_{\text{modeled}}$, was computed and normalized ($\varepsilon_{\text{normalized}} = \frac{\varepsilon_{\text{modeled}} - \varepsilon_{\text{res}}}{\varepsilon_{\text{int}} - \varepsilon_{\text{res}}}$) yielding values between 0 (fully frozen) and 1 (fully unfrozen). The degree of soil being unfrozen was determined as the mean of $\varepsilon_{\text{normalized}}$ across all $N = 15,000$ simulations, given by $\frac{1}{N} \sum_{i=1}^{N} \varepsilon_{\text{normalized},i}$. Consequently, the degree of soil freezing, which can be interpreted as the probability of frozen ground, was expressed as $1 - \frac{1}{N} \sum_{i=1}^{N} \varepsilon_{\text{normalized},i}$.

Because satellite overpasses motivate a daily product, we aggregated the hourly probabilities to daily means. Daily soil state was then labelled as frozen if the mean freeze probability exceeded 0.9, unfrozen if it was below 0.1, and in transition (partially frozen) otherwise. These thresholds were selected to ensure high confidence in identifying predominantly frozen or unfrozen conditions, while intermediate values captured transitional states associated with partial freezing.



### 2.5.2 Temperature-based Soil State Classification

The conventional hard $0\,^\circ$C threshold was modified by introducing a probabilistic analogue that accounted for sensor uncertainty. Each hourly soil temperature observation was sampled 1,000 times from a normal distribution, $T_{\text{sim}} \sim \mathcal{N}(T_{\text{obs}}, \sigma_T^2)$, where $\sigma_T$ represented the sensor-specific standard deviation. These samples were then passed through a logistic function, $P_{\text{frozen}}(T) = \frac{1}{1+\exp[s(T-T_0)]}$, with $T_0 = 0\,^\circ$C and a steepness parameter $s = 5$, to compute a smoothed freeze probability. Daily mean probabilities were subsequently used to classify the soil as frozen if $P_{\text{frozen}} > 0.5$, and unfrozen otherwise. Because this
temperature-based method could not resolve a transitional state, a neutral threshold of 0.5 was applied, and transitional days were therefore not explicitly represented in this scheme.

## 3   Results

### 3.1   Model Parameter Analysis

A total of 204 freezing cycles were identified across all sites, and the SFCC model (Eq. 4) was fitted to each cycle separately.
Since a mean $R^2$ below 0.95 indicates a poor fit, cycles with an $R^2$ value below 0.6 were excluded (4 cycles) for quality control. Additionally, 11 cycles were excluded due to unreliable parameter estimates as evidenced by frequent boundary values for $T_f$, $T_{\text{res}}$, and $b$ during bootstrapping. To further ensure robustness, we applied an interquartile range (IQR)-based filter to exclude cycles with excessively wide confidence intervals, defined as greater than $Q3 + 3 \times \text{IQR}$ for each parameter separately, resulting in the removal of 35 cycles. A final visual inspection led to the exclusion of 8 additional cycles identified with anomalies, such
as irregular water content changes during freezing. Following this rigorous filtering process, 146 freezing cycles were retained: 12 from tundra sites, 19 from boreal forest sites (8 from eastern boreal networks and 11 from western boreal networks), and 115 from prairie sites.

The mean values, standard deviations across cycles ($\sigma$), and mean bootstrapped standard deviations ($\overline{\sigma_{\text{bootstrap}}}$) for each parameter were calculated to assess the freezing characteristics of soils in our networks across different landscapes. It is important
to note that these networks may not be fully representative of the entire landcover but rather indicate strong spatial variability in freezing processes. The results of this section are summarized in Table 2. In the eastern boreal forest, the mean $b$ was 3.41 ($\sigma = 0.98$, $\overline{\sigma_{\text{bootstrap}}} = 1.08$), the highest among all networks, suggesting a sharp freezing transition. In contrast, the western boreal forest had the lowest mean $b$ at 1.33 ($\sigma = 0.87$, $\overline{\sigma_{\text{bootstrap}}} = 0.21$), indicating a more gradual freezing transition. The lowest $T_{\text{res}}$ was observed in the prairies at $-4.72\,^\circ$C ($\sigma = 3.18\,^\circ$C, $\overline{\sigma_{\text{bootstrap}}} = 0.48\,^\circ$C), suggesting that prairie soils reach a completely
frozen state at lower temperatures than other landcovers. In contrast, the eastern boreal forest exhibited a $T_{\text{res}}$ of $-1.07\,^\circ$C, indicating that soils in this region freeze fully at relatively higher temperatures. The highest freezing onset temperature ($T_f$) was recorded in the eastern boreal forest at $0.47\,^\circ$C ($\sigma = 0.20\,^\circ$C, $\overline{\sigma_{\text{bootstrap}}} = 0.07\,^\circ$C), suggesting that soils in this region begin freezing at a higher temperature. Conversely, the western boreal forest exhibited a more depressed freezing temperature.

To further investigate the model parameters, we analyzed the distribution of $b$, $T_f$, and $T_{\text{res}}$ values (Fig. 3). The analysis
reveals that extreme cases in the boxplots are rare, as expected, and cannot be classified as outliers due to the rigorous filtering





**Table 2.** Summary of SFCC model parameters by land cover

| Land cover | n | Mean $b$ ($\sigma$) | Mean $T_f$ ($\sigma$) | Mean $T_{\text{res}}$ ($\sigma$) |
|---|---|---|---|---|
| Eastern boreal forest | 8 | 3.41 (0.98) | 0.47°C (0.20°C) | -1.07°C (0.43°C) |
| Western boreal forest | 11 | 1.33 (0.87) | 0.00°C (0.67°C) | -3.32°C (1.77°C) |
| Prairie | 115 | 1.63 (1.53) | 0.08°C (0.27°C) | -4.72°C (3.18°C) |
| Tundra | 12 | 2.6 (1.56) | 0.13°C (0.23°C) | -2.45°C (1.56°C) |

applied to the cycles. The extreme $b$ values observed in the prairie region are primarily associated with cycles recorded during spring. During this period, abundant water from snowmelt, combined with the absence of snow cover, allows diurnal air temperature fluctuations to drive rapid freezing cycles, resulting in elevated $b$ values. For instance, the two cycles with $b$ values greater than 6 correspond to $\varepsilon_{\text{int}}$ values of approximately 36, significantly higher than the network average of 16.5. In the prairie
region, the combination of high clay content and low moisture content prior to freezing generally leads to higher $T_{\text{res}}$ values. Notably, the site with the lowest recorded $T_{\text{res}}$ (UG17, -14.94°C) also exhibited the highest clay content (41.1%; (Tetlock et al., 2019)) among all sites. In the western boreal forest, one extreme $T_f$ value (-1.63°C) was observed at a site within the Candle Lake network, where the moisture content was exceptionally low. The $\varepsilon_{\text{int}}$ for this site was approximately 3.57, the lowest among the boreal forest sites, which contributed to the pronounced depression in $T_f$.

## 3.2 Model Performance

We evaluated the reliability of the constructed SFCC in estimating the degree of soil freezing and classifying soil states by comparing SFCC-based predictions of $\varepsilon_{\text{eff}_{\text{Predicted}}}$—obtained via Monte Carlo simulations—with in situ $\varepsilon_{\text{eff}_{\text{Actual}}}$ measurements (Fig. 4). Model performance was quantified using the coefficient of determination ($R^2$), root mean square error (RMSE), and mean absolute error (MAE). Additionally, relative RMSE and MAE were calculated as percentages of the mean $\varepsilon_{\text{eff}_{\text{Actual}}}$
(Table 3). Overall, the model demonstrated strong performance in predicting $\varepsilon_{\text{eff}}$. However, high relative errors were observed in tundra regions, likely due to multiple incomplete freezing and thawing cycles preceding the main freezing cycle. These fluctuations introduce uncertainties in the transitional zone, making it more challenging for the model to precisely predict $\varepsilon_{\text{eff}}$.

**Table 3.** Model performance metrics for predicting $\varepsilon_{\text{eff}}$ across different land cover types during freezing periods. Relative errors are shown in parentheses as percentages of the mean $\varepsilon_{\text{eff}_{\text{Actual}}}$.

| Land Cover Type | $R^2$ | RMSE | MAE |
|---|---|---|---|
| Eastern boreal forest | 0.85 | 1.25 (18.28%) | 0.81 (11.85%) |
| Western boreal forest | 0.94 | 0.29 (8.03%) | 0.18 (4.99%) |
| Prairies | 0.95 | 1.07 (10.94%) | 0.65 (6.65%) |
| Tundra | 0.86 | 1.71 (39.20%) | 0.87 (19.94%) |
| Overall | 0.95 | 1.16 (14.58%) | 0.64 (8.05%) |





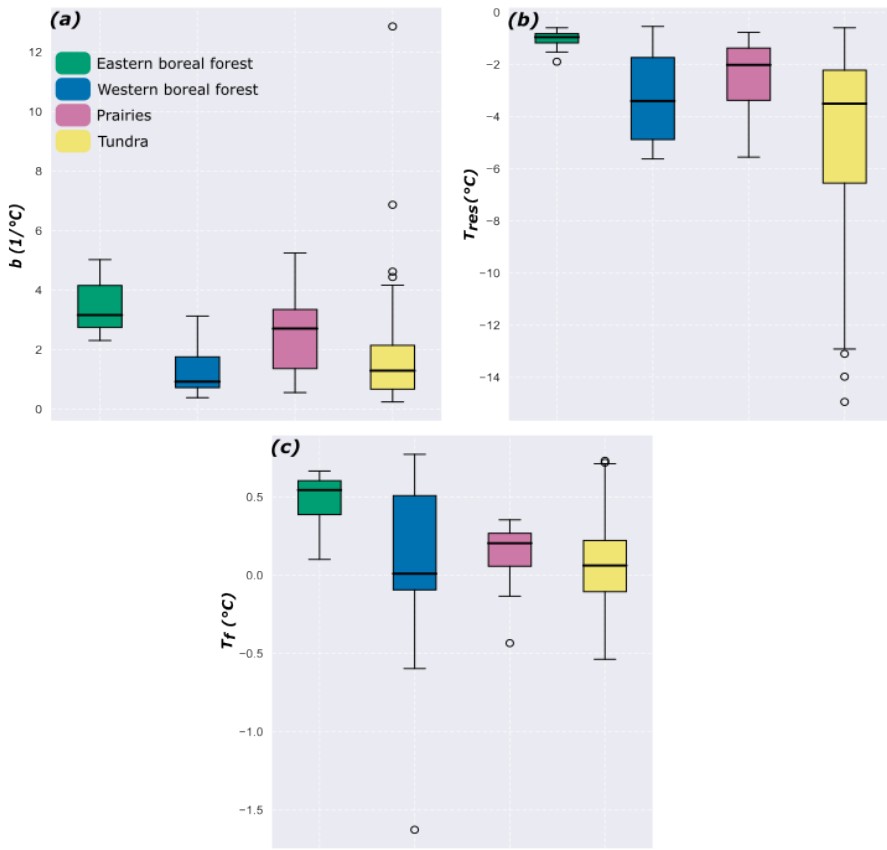

**Figure 3.** Box plots showing the distribution of model parameters: shape factor $b$ **(a)**, $T_{\text{res}}$ **(b)**, and $T_f$ **(c)** across different land cover types.

Figure 5 presents the cumulative distribution functions (CDFs) of the residuals, highlighting key patterns in the model's performance across different landcover types and illustrating the distribution of residuals as a function of soil temperature.

Across all regions, residuals are smallest at temperatures well below freezing, demonstrating the model's strong ability to predict soil effective permittivity accurately under stable frozen conditions where variability in total water content is minimal. Conversely, residuals increase significantly near the freezing point, where abrupt changes in soil temperature and water content introduce greater variability, challenging the model's predictive accuracy. At temperatures above freezing, residuals remain larger compared to subfreezing conditions, likely due to increased fluctuations in total water content caused by rainfall events,

as expected. The CDFs exhibit steep slopes near zero across all landcover types, indicating minimal systematic bias in the model and confirming that the majority of residuals are small, thereby supporting the model's overall robustness. However, the eastern boreal forest, prairie, and tundra regions exhibit pronounced tails in their residual distributions, indicating instances of underprediction and overprediction. In contrast, the western boreal forest demonstrates the most accurate predictions, as evidenced by its steep CDF near zero and minimal tail lengths. This superior performance may be attributed to the region's

drier conditions and smaller fluctuations in total water content, which enhance model reliability.

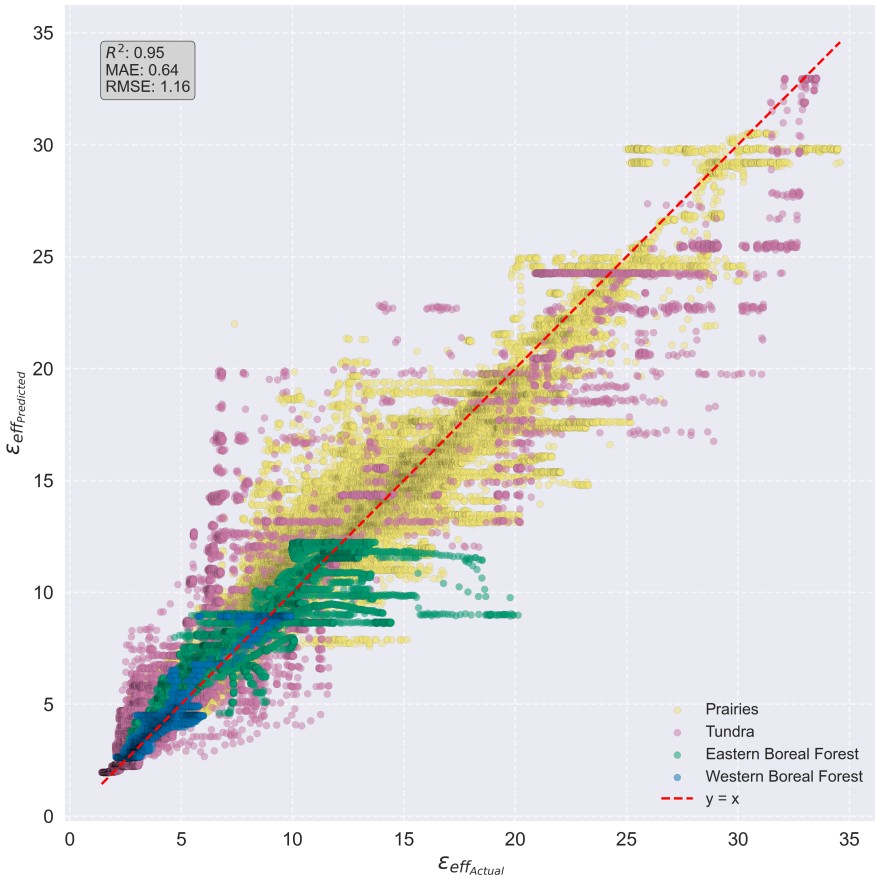

**Figure 4.** Comparison of in situ (actual) and SFCC-predicted $\varepsilon_{\mathrm{eff}}$ for freezing cycles, color-coded by land cover type.

We also tested the model's performance during thawing cycles, both for the full thawing period and before snowmelt (or any water infiltration into the soil), to evaluate our assumption that the SFCC is equivalent to the STCC. The model performed well in estimating $\varepsilon_{\mathrm{eff}}$ before snowmelt. Across land cover types, the coefficient of determination ($R^2$) values were 0.54 for the eastern boreal forest sites, 0.70 for the western boreal forest sites, and 0.65 for the prairie sites. RMSE values were

0.84 (18.00%), 0.13 (4.43%), and 1.46 (19.65%), while MAE values were 0.44 (9.56%), 0.10 (3.36%), and 0.80 (10.79%), respectively. (Relative errors, expressed as percentages of the mean $\varepsilon_{\mathrm{eff}}$, are shown in parentheses.) Including data from the post-snowmelt period reduced the $R^2$ values to 0.04 in the eastern boreal forest sites, 0.16 in the western boreal forest sites, and 0.54 in the prairies sites. RMSE values increased to 6.07 (69.60%), 3.76 (83.95%), and 4.82 (46.65%), while MAE values rose to 3.36 (38.48%), 1.23 (27.35%), and 2.21 (21.42%), respectively. These results clearly indicate that after the onset of

snowmelt, the model's performance declined significantly, particularly in boreal forest regions. This decline can be attributed to substantial snow cover, which led to an influx of water, causing the total soil water content to far exceed its initial pre-freezing levels. In contrast, the inclusion of post-snowmelt data had minimal impact on the prairies, likely due to the region's limited





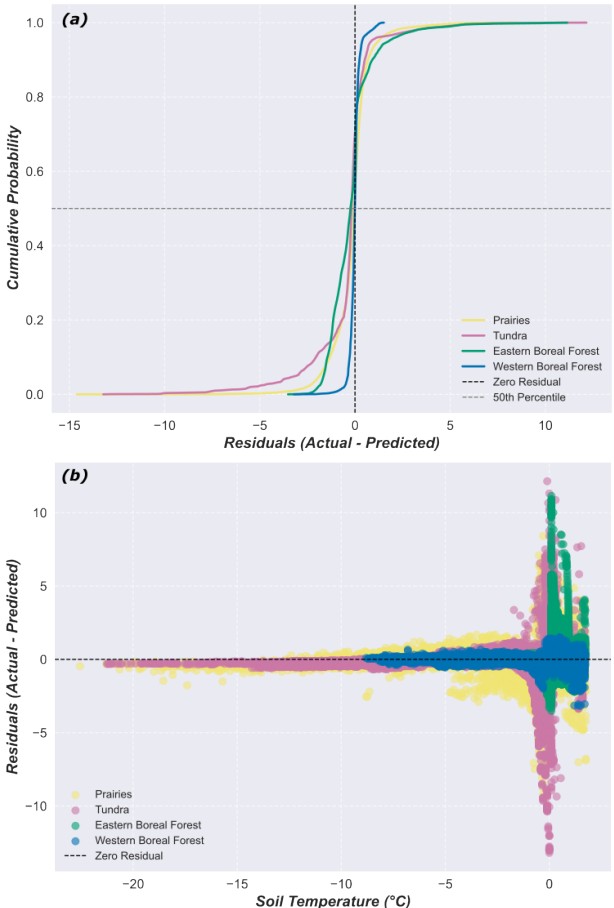

**Figure 5.** CDFs of residuals across landcover types (**a**), showing that most residuals are small, with steep slopes near zero indicating minimal systematic bias. Residuals as a function of soil temperature (**b**), with an equal number of data points randomly selected per landcover to ensure fair representation.

snow cover, which resulted in minimal additional water input during thawing. In summary, the model effectively predicted $\varepsilon_{\mathrm{eff}}$ during the early thawing phase, supporting the assumption that SFCC $\approx$ STCC. However, this assumption became invalid after the onset of the snowmelt, particularly in boreal forests, where the large influx of meltwater significantly altered soil water content.

## 3.3 Model Application to Field Data

To illustrate the application of our SFCC model (Eq. 4) and its integration with in situ data, we presented five example sites from different monitoring networks, each exhibiting distinct freezing behaviors: eastern boreal forest (Fig. 6 and Fig. 7), western boreal forest (Fig. 8), prairie (Fig. 10), and tundra (Fig. 9). Each example consisted of two panels: the first panel (a)



depicted the fitted SFCC overlaid on the processed in situ measurements of soil temperature and permittivity, with vertical lines indicating $T_f$ and $T_{res}$. The second panel (b) displayed the time series of soil temperature for the same site, color-coded by the probability of frozen ground (degree of soil freezing). To further summarize the results, we included Fig. 11, which shows the monthly average of the probability of frozen ground over the entire data period for all sites within each network. This visualization provided insight into the average freezing and thawing patterns at each station and network. In the eastern boreal forest, soils rarely freeze completely during winter (Fig. 11), reflecting high water retention and the insulating effect of snow and vegetation cover. This leads to either prolonged transitional states—a phenomenon known as the zero curtain effect—or unfrozen soil throughout the year. On average, we recorded 23 frozen days and 46 transitional days in this region. In contrast, the western boreal forest, characterized by drier conditions compared to its eastern counterpart, experiences more extensive freezing while still retaining some transitional states. On average, we recorded 73 frozen days and 76 transitional days at these sites. As expected, tundra sites exhibited the longest frozen periods, with an average of 145 frozen days due to consistently low temperatures, along with 52 transitional days. Prairie soils began freezing earlier than boreal forest soils, likely due to the absence or shallow depth of snow cover and the lack of vegetation, as these sites are primarily agricultural lands. This lack of insulation made prairie soils more susceptible to air temperature fluctuations (Fig 10), allowing soil temperatures to drop more rapidly. On average, we recorded 71 frozen days and 71 transitional days in the prairies. Additionally, prairie soils thaw earlier than other landcover types, reflecting their sensitivity to air temperature variations.

### 3.4 SFCC-Based Soil Classification vs. Temperature-Based Classification

The agreement between the SFCC-based classification and the temperature-based method varied systematically across different land cover types (Fig. 12). In the eastern boreal forest, the overall agreement was 38.0%. Approximately 86% of the transitional days identified by the SFCC method were mislabeled as "unfrozen," leading to a substantial overestimation of unfrozen conditions. In the western boreal forest, agreement was 54.1%; here, nearly 87% of transitional days were misclassified as "frozen," resulting in an overestimation of frozen ground. The prairies showed 67.2% agreement, with a similar pattern—about 94% of transitional days were labeled as "frozen," again inflating the estimated extent of frozen conditions. The tundra exhibited the highest agreement at 75.0%, likely due to a shorter transitional phase; in this region, around 64% of transitional days were classified as "unfrozen" and 36% as "frozen," leading to smaller and more balanced misclassification. This analysis confirms that the temperature-based method systematically reallocates transitional days into binary categories, and that the direction and magnitude of these biases vary across ecological regions due to differing soil and climate conditions.

## 4 Discussion

The presence and behavior of liquid water, and consequently the shape of the SFCC, are influenced by numerous interrelated factors, including soil mineral composition, particle size, plasticity, initial water content, dry density, solute concentration, freezing rate, confining stress, and freeze-thaw history (Chai et al., 2018; Bi et al., 2023a, b, and references therein). The freezing point ($T_f$) decreases as initial water content decreases. For example, Zhang et al. (2019) found that decreasing soil

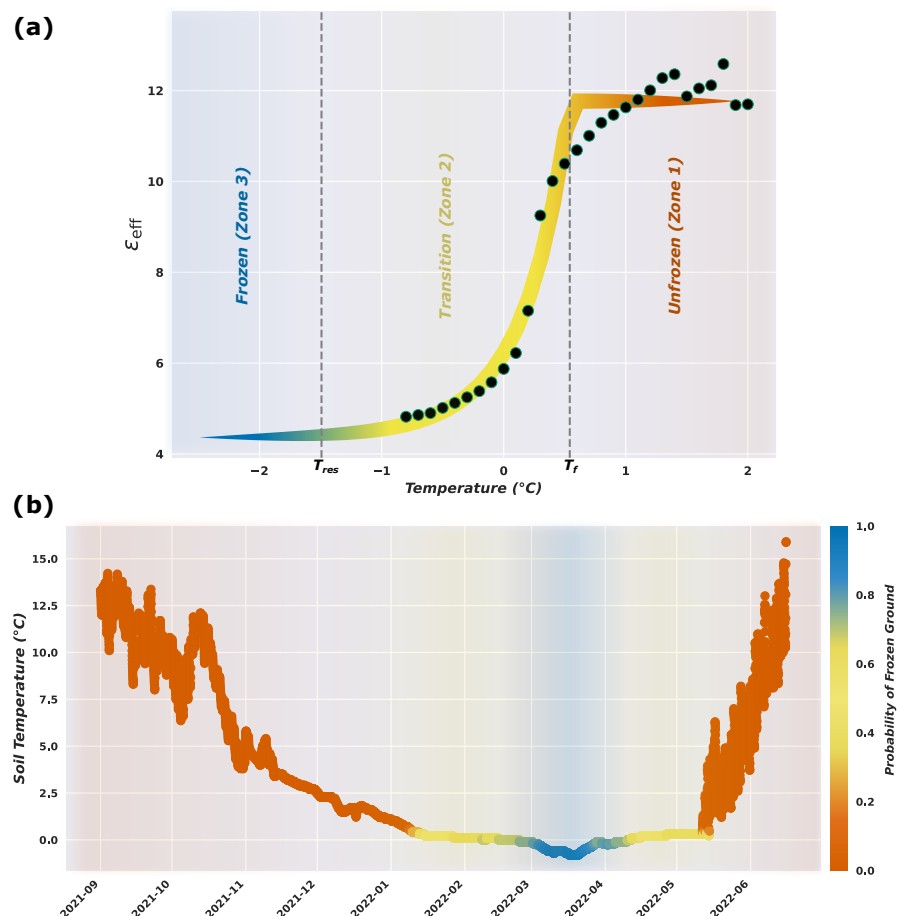

**Figure 6.** Fitted SFCC (a) and time series of soil temperature (b), color-coded by the probability of frozen ground, for the BJ06 site in the Baie James network, an eastern boreal forest site. Vertical lines in (a) indicate $T_f$ (freezing point) and $T_{\mathrm{res}}$ (residual temperature). The prolonged transitional states highlight the zero curtain effect. The site's location can be found through an interactive map hosted on GitHub: Soil-Temperature-Permittivity-Monitoring-Sites.

water content from 37.60% to 14.52% lowered $T_f$ from $-0.06°$C to $-0.75°$C. A similar pattern emerges in our results, where eastern boreal forest sites, characterized by higher initial moisture (indicated by higher $\varepsilon_{\mathrm{int}}$), display higher $T_f$ than the drier
western boreal sites. Soil texture also influences the SFCC. Fine-grained soils show greater freezing point depression and gradual changes in liquid water content, while coarse-grained soils exhibit rapid water loss during freezing (Tian et al., 2014; Zhang et al., 2019; Bi et al., 2023b). This contrast is evident at the Kenaston network, where higher clay content correlates with lower $T_{\mathrm{res}}$ and a more gradual freezing transition. Additionally, the freezing rate, by altering pore ice formation dynamics, solute distribution, and ice grain size, can affect both the slope and curvature of the SFCC (Watanabe and Osada, 2017). In a natural
setting, the freezing rate is primarily driven by decreases in ambient (air) temperature—much like in controlled laboratory




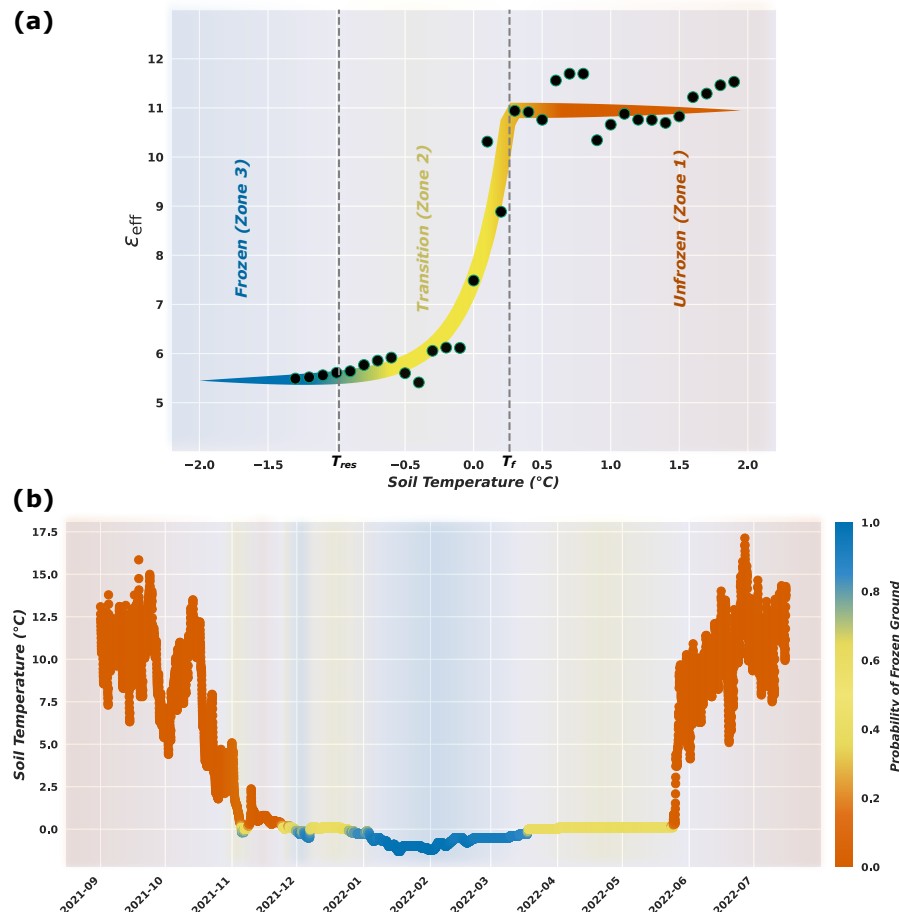

**Figure 7.** Similar to Fig. 6, but for the FM403 site in the Montmorency Forest network, an eastern boreal forest site located at the transition between boreal and temperate forests. The site experienced two freeze-thaw events during the year, as captured by our SFCC model, with prolonged transitional states reflecting the zero curtain effect.

conditions—yet it is further modulated by the presence of vegetation and snow cover, thus complicating efforts to disentangle the specific effects of air temperature on the SFCC. For instance, in eastern boreal forest sites, abundant snow and insulating moss-lichen cover prolong the zero-curtain period, allowing soil temperatures to remain near the freezing point even when air temperatures plummet below $-30°$C. Despite recognizing these controlling factors, fully disentangling their individual
contributions within a natural, open-system environment is inherently challenging. Multiple drivers operate simultaneously and interact nonlinearly, making it difficult to isolate each parameter's effect. Crucially, however, our primary objective was not to develop a universal SFCC model. Instead, we aimed to demonstrate how SFCCs constructed in permittivity–temperature space can enhance the understanding of soil freeze-thaw behavior under real-world conditions. Attempting to explain every observed SFCC from in situ measurements alone is neither feasible nor necessary. Instead, our approach involves fitting individual




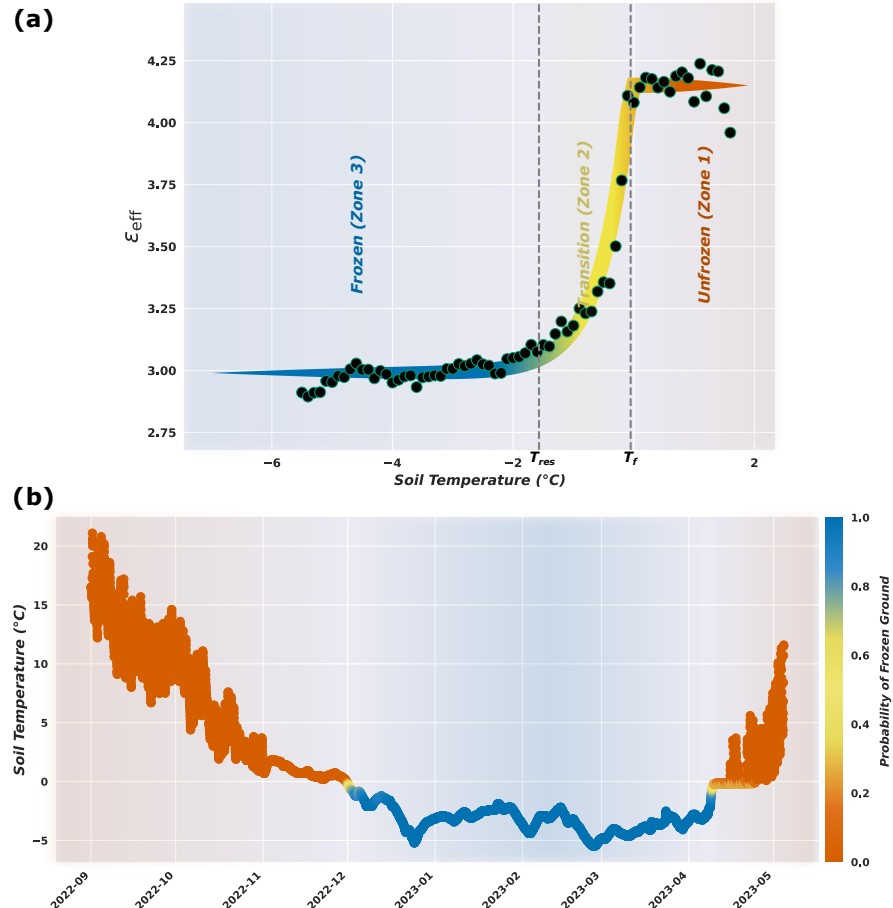

**Figure 8.** Similar to Fig. 6, but for the BT17 site in the Candle Lake network, a western boreal forest site. The more extensive freezing reflects drier conditions and lack of insulation compared to eastern sites, with some transitional states present.

SFCCs to each freeze-thaw event, embracing their site- and event-specific nature. This strategy avoids the pitfalls of using averaged parameters (such as those in Table 2) or assuming a one-size-fits-all model. It's important to note that the averaged parameters presented in Table 2 should not be used as universal values to determine soil state from temperature data alone. While we can identify general trends and influencing factors, each SFCC must ultimately be interpreted within its unique environmental and soil context.

One key limitation of this study, as well as any attempt to construct SFCCs from in situ measurements, lies in the assumption that the total water content remains equal to the pre-freezing water content throughout the freezing period (He and Dyck, 2013). In an open system, soil moisture can fluctuate at any time due to rainfall events, intermittent snowmelt, other hydrological inputs or water redistribution during freezing. This particularly makes identifying a stable "initial" water content (or corresponding $\varepsilon_{eff}$) challenging. However, we found that averaging the permittivity within a small temperature range





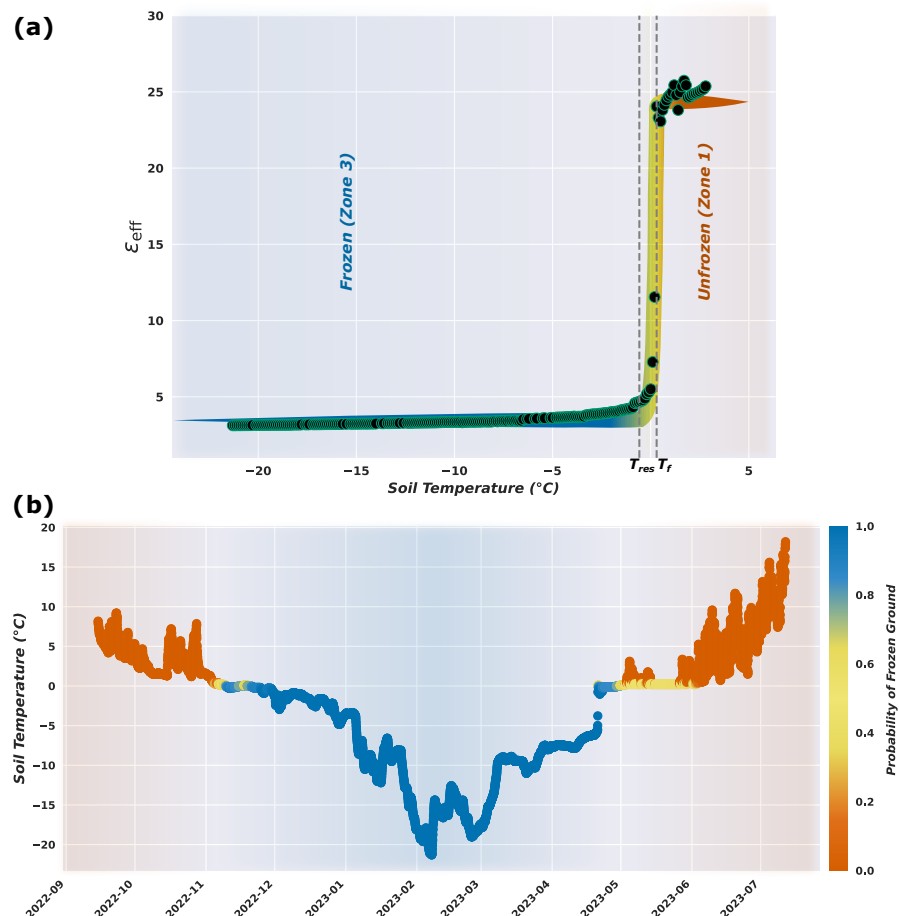

**Figure 9.** Similar to Fig. 6, but for the GR01 site in the George River network, a tundra site. Sharp and rapid freezing transitions result in minimal transitional periods.

(e.g., $\sigma_T$°C to 2°C, where $\sigma_T$ represents the instrument-specific temperature uncertainty) before the onset of sustained cooling provides a reasonable proxy for the initial moisture conditions. Nevertheless, our results indicate that under freezing conditions—before major thawing or significant water inputs—this assumption holds reasonably well, allowing us to accurately reconstruct $\varepsilon_{\text{eff}}$ from the fitted SFCC. This, in turn, lends credibility to using the derived SFCC parameters to define soil states. Although this approach may introduce some uncertainties, it is both practical and effective under most natural conditions. Inte-

grating air temperature measurements can help address certain complexities. For instance, a concurrent rise in air temperature above zero and an increase in permittivity may indicate external water inputs (e.g., snowmelt) that violate our assumption. Another practical challenge in applying the SFCC approach in situ is identifying distinct freezing and thawing periods. In controlled laboratory settings, these phases are straightforward to define due to precisely managed temperature profiles. In natural environments, however, air temperatures fluctuate continuously—often with pronounced diurnal cycles—leading to brief or in-



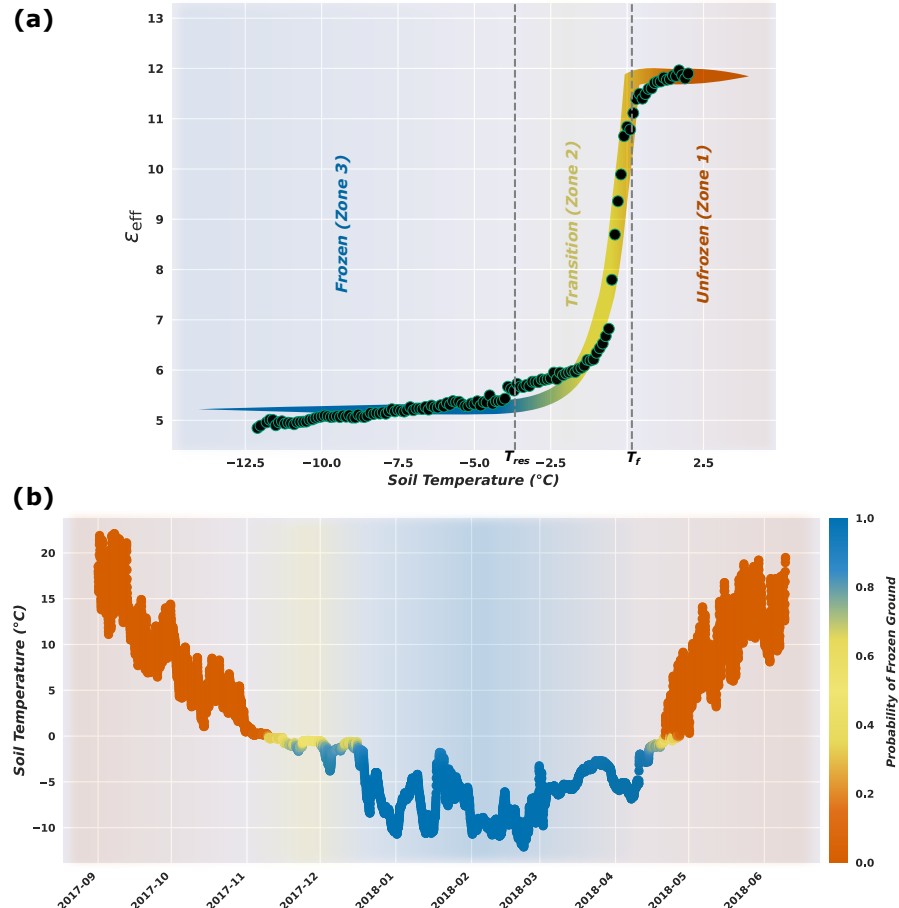

**Figure 10.** Similar to Fig. 6, but for the EC17 site in the Kenaston network, a prairie site. Pronounced fluctuations in soil temperature are observed, likely due to minimal snow cover and absence of vegetation, which reduce insulation and increase susceptibility to air temperature fluctuations.

complete freeze-thaw events that are difficult to isolate. These fluctuations are particularly common during the fall and spring shoulder seasons, precisely when soils transition between unfrozen and frozen states. Notably, these transitional periods also provide the critical data needed for deriving $T_f$ and $b$, making SFCC construction even more challenging. For practical purposes and to improve the reliability of SFCC fitting, we recommend focusing on the main, more sustained freezing and thawing periods, while disregarding minor, short-lived fluctuations near $0°C$. Although this approach may exclude some small-scale

freeze-thaw cycles, prioritizing the most clearly defined freeze-thaw phases balances the complexities of natural systems with the need for practical, reliable SFCC parameter estimation.

For seasonally frozen ground monitoring, sensors that measure both soil temperature and permittivity are essential. The HydraProbe is advantageous as it directly measures permittivity components, while the TEROS-12, despite requiring empirical





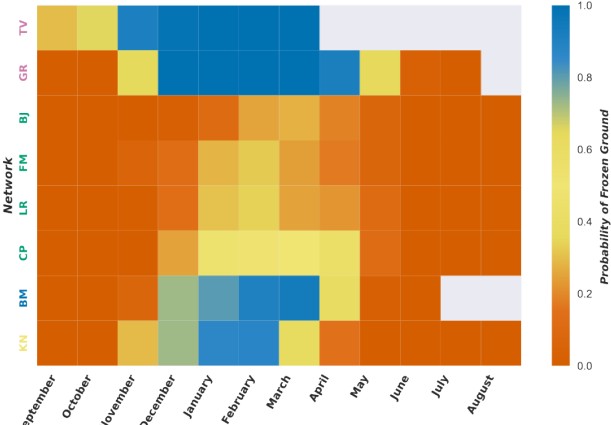

**Figure 11.** Heatmap showing the monthly average of the probability of frozen ground for all sites within each network over the entire data period. This visualization highlights regional freeze-thaw patterns: tundra remains frozen longest, prairies freeze earlier but thaw faster due to minimal snow and vegetation, and the eastern boreal forest retains transitional states, indicating incomplete freezing from high water retention and snow insulation.

conversion, offers exceptional energy efficiency. The CS616 is less suitable due to its lack of integrated temperature mea-
surement, undefined permittivity conversion, and poor reliability in cold conditions observed at our Chapleau site. We did not
perform site- or sensor-specific calibrations for permittivity or temperature, but the sensors used—TEROS 12, HydraProbe, and
CS616—have been extensively validated and shown to perform reliably across diverse soil conditions (Seyfried and Murdock,
2004; Kelleners et al., 2005; Hansson and Lundin, 2006; Cominelli et al., 2024; Pardo Lara et al., 2021). Electrical conduc-
tivity (EC) is the most influential factor affecting dielectric sensor accuracy. Increased EC elevates the imaginary permittivity
component, leading to signal attenuation and overestimated permittivity values (Seyfried et al., 2005). However, EC values at
our sites were generally low—below 0.03 S m$^{-1}$ in boreal and tundra regions (Fig. S1) and below 0.2 S m$^{-1}$ in Kenaston's
top 20 cm (Tetlock et al., 2019)—well within acceptable thresholds (0.05–0.14 S m$^{-1}$ depending on the sensor). Permittivity
for TEROS 12 and CS616 was derived from raw sensor output using manufacturer-recommended or physically based models.
For CS616, we applied the formulation by Kelleners et al. (2005), using generalized calibration coefficients from prior studies
(Kelleners et al., 2005; Logsdon, 2009; Hansson and Lundin, 2006). While this may introduce minor biases, the impact on
freeze–thaw detection is negligible. For TEROS 12, we used a third-order polynomial to convert frequency to permittivity
(see Equation A2), though underestimation has been reported in saturated conditions (Cominelli et al., 2024; Fragkos et al.,
2024). This is unlikely to affect our analysis, as saturated soils are rare during freezing periods—except at Montmorency,
where soil rarely freezes. Importantly, SFCCs can also be constructed directly from raw sensor output, with negligible differ-
ences compared to permittivity-based curves (Fig. S2). Both approaches reveal the clear signal drop needed to identify freezing
transitions. Soil texture and organic matter may also affect permittivity measurements (Seyfried and Murdock, 2004; Seyfried
et al., 2005), but most sites exhibit low clay and organic content, with only a few exceptions in Kenaston and Montmorency.



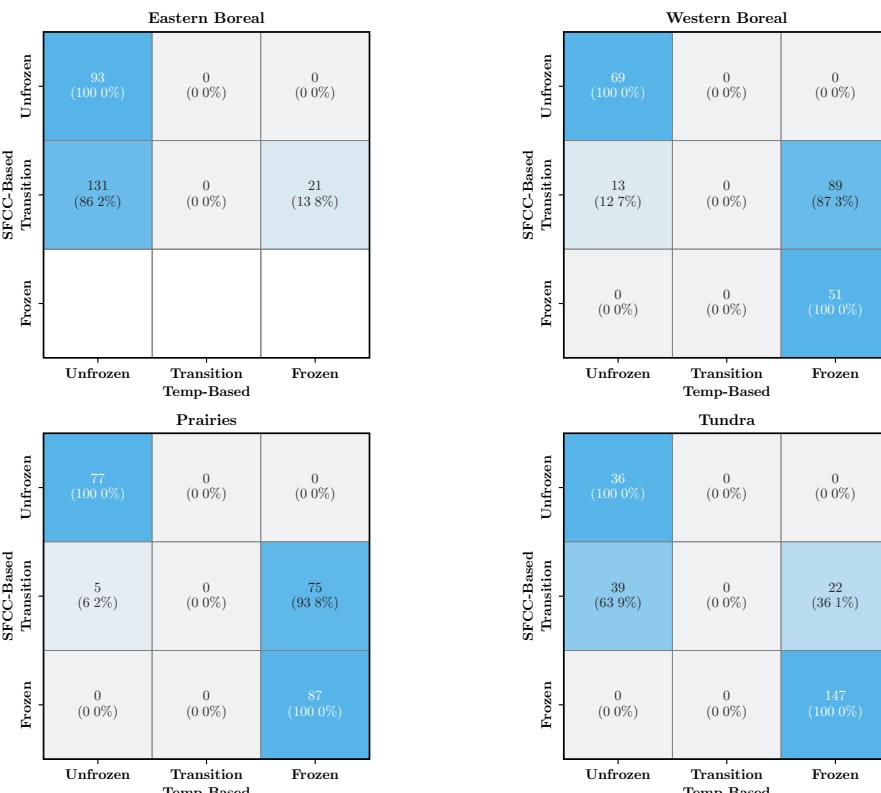

**Figure 12.** Confusion matrix comparing SFCC-based soil state classification (used as the reference) with the temperature-based method across different networks in various land cover types. The analysis is based on daily classifications from October 1st to June 1st.

Overall, the permittivity uncertainty is approximately 1–2 units for CS616 and TEROS 12, and about 0.1–0.2 units for the HydraProbe—values that are well below the typical permittivity shifts observed during soil freezing and thawing. As long

as a discernible permittivity change occurs, the SFCC fitting method can reliably identify the freeze–thaw transition. In rare instances where the change is too subtle—e.g., below the measurement uncertainty threshold—the curve-fitting algorithm fails to converge, and such data are automatically excluded from further analysis. Importantly, our detection approach is based on the relative change in permittivity with temperature rather than the absolute permittivity values, meaning that minor calibration errors or site-specific variability do not compromise our ability to detect meaningful freeze–thaw events.

A major source of systematic error in permittivity-based sensors is the volume mismatch between the permittivity-sensing domain and the temperature-sensing thermistor (Pardo Lara et al., 2020, 2021). As shown by Pardo Lara et al. (2021), this mismatch can lead to apparent hysteresis and positive freezing-point depression artifacts, where permittivity sensors detect freezing before the thermistor. This occurs because permittivity sensors integrate over a larger, water-biased volume—one that shrinks in wetter soils—while thermistors measure temperature within a much smaller, localized, and water-independent zone

(Hansson and Lundin, 2006; Logsdon, 2009; Pardo Lara et al., 2021). Sensor geometry also plays a role. The thermistor in the





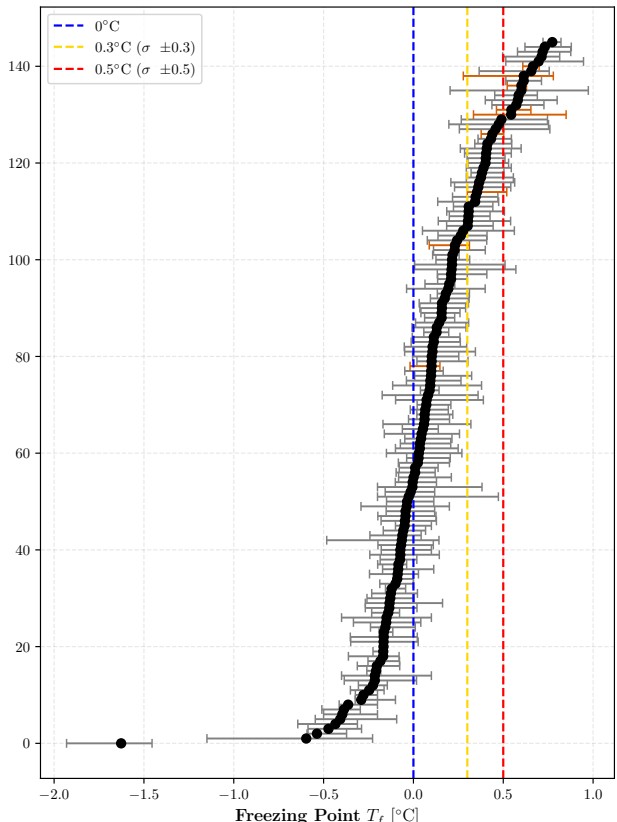

**Figure 13.** Bootstrapped estimates of $T_f$ and their 95% confidence intervals for individual freezing cycles. Cycles from sensors with higher temperature uncertainty (0.5°C; e.g., TEROS 12 and CS616) are shown in red, while those from lower uncertainty sensors (0.3°C; e.g., HydraProbe) are shown in grey. Dashed vertical lines at 0.3°C and 0.5°C represent one standard deviation ($\sigma$) of the expected uncertainty bounds for each sensor type.

TEROS 12 is embedded in the central needle, minimizing spatial offset, while in the HydraProbe it is located in the base plate. The CS616 lacks a built-in thermistor, requiring external placement, which can exacerbate mismatch effects. In our analysis (Fig. 13), approximately 75% of $T_f$ values from HydraProbe and 50% from TEROS 12 and CS616 were within the respective sensor uncertainty ranges ($\pm 0.3\,°$C and $\pm 0.5\,°$C). Rather than dismissing these positive $T_f$ values as sensor error, we interpret
them as systematic biases resulting from volume mismatch. As they still reflect the relative permittivity change associated with freezing, we consider them informative and include them in our analysis.

By comparing the number of frozen days determined using only soil temperature measurements with those derived from our SFCC-based method, we found that the traditional threshold-based approach substantially overestimates the duration of frozen ground. This result underscores the importance of revisiting conventional monitoring practices and reaffirms that proper
frozen ground monitoring sites should include soil moisture sensors. Furthermore, relying solely on temperature fails to capture





extensive transitional periods, which can be comparable in length to the fully frozen intervals. In light of these findings, we recommend moving away from the notion of a single 'frozen period' defined by consecutive frozen days, and instead adopting a more detailed approach that quantifies the number of individual frozen, transitional, and unfrozen days. Such a perspective provides a dynamic and accurate representation of soil freeze-thaw behavior, acknowledging the frequent transitions and short-term fluctuations that occur in natural settings. This finer-scale approach more closely aligns with the actual processes taking place in the soil, allowing for improved monitoring and understanding of seasonally frozen ground conditions.

## 5 Conclusions

This study introduces a new framework for seasonally frozen ground monitoring that moves beyond the traditional binary classification approach, offering a more nuanced understanding of soil freeze-thaw dynamics. By integrating in situ soil temperature and permittivity measurements through an SFCC model, we can now quantify the degree of soil freezing and identify crucial transitional states—a capability particularly valuable for understanding shoulder season processes and carbon flux dynamics. This advancement addresses a critical gap in cold regions science, where accurate characterization of soil states directly impacts our understanding of hydrological processes, ecosystem responses, and carbon cycling. The framework's ability to detect and quantify transitional states, which we found can persist as long as fully frozen periods, has significant implications for improving climate models, particularly their representation of shoulder season biogeochemical processes. Additionally, this methodology provides a robust foundation for validating and improving remote sensing products, potentially enabling more accurate regional and global assessments of frozen ground conditions. Looking forward, this approach opens new avenues for integrating ground-based observations with satellite data, ultimately advancing our ability to monitor and predict cold region responses to climate change.

## Appendix A: Sensor Details

### A1 CS616 (Campbell Scientific)

The CS616 water content reflectometer measures soil dielectric permittivity by recording the period (in microseconds) of a square-wave oscillation. This oscillation is generated by an electromagnetic pulse that travels along the sensor's 30 cm stainless steel rods, reflects off their ends, and returns to the circuit board to trigger the next pulse. Since the wave velocity depends on the dielectric properties of the surrounding medium, the measured period is directly related to the effective relative permittivity ($\varepsilon_{\text{eff}}$). A physically based equation derived by Kelleners et al. (2005) can be used to convert the raw output to $\varepsilon_{\text{eff}}$:

$$\varepsilon_{\text{eff}} = \left( \frac{(t - 2t_d)c}{4L} \right)^2 \tag{A1}$$

where $t = \tau/S_t$ is the scaled time period (with $S_t = 1024$ and $\tau$ the temperature-corrected raw output in seconds), $t_d$ is the delay time correction (commonly $5.4 \times 10^{-9}$ s), $L$ is the effective rod length (typically 0.261 m for the CS616), and $c$ is the



speed of light in a vacuum ($3 \times 10^8$ m/s) (Hansson and Lundin, 2006; Kelleners et al., 2005; Logsdon, 2009). Although $L$ and $t_d$ can vary slightly between CS616 probes, studies have shown minimal variability (Kelleners et al., 2005; Hansson and Lundin, 2006; Logsdon, 2009). These generalized constants yield acceptable permittivity estimates when sensor-specific calibration is not feasible. Validation against standard reference fluids showed excellent agreement ($R^2 > 0.99$), and comparisons with TDR, HydraProbe, and Topp's model confirm reliable performance across soils. The accuracy of the CS616 may decline in soils with

high EC ($>0.05\,\mathrm{S\,m^{-1}}$), clay content ($>30\,\%$), or organic matter ($>5\,\%$) due to signal attenuation and delayed pulse detection. However, the CS616's relatively high operating frequency ($\sim 175\,\mathrm{MHz}$) reduces sensitivity to dispersive effects. The CS616's sensing volume averages permittivity over a non-uniform electric field, which is biased toward wetter zones. While this may slightly inflate permittivity in heterogeneous soils, it helps reduce the influence of small-scale spatial variability. While the CS616 does not measure soil temperature directly, it can be paired with the CS109SS-L sensor for temperature measurements.

The CS109SS-L operates over a temperature range of $-40^\circ$C to $+70^\circ$C, with an accuracy of $\pm 0.60^\circ$C from $-40^\circ$C to $-20^\circ$C and $\pm 0.49^\circ$C from $-20^\circ$C to $+70^\circ$C.

## A2  TEROS-12 (METER Group)

The TEROS 12 (METER Group, Inc.) uses capacitance-based technology to estimate soil dielectric permittivity. Operating at 70 MHz, it sends an oscillating signal through three 5.5 cm prongs, which act as a capacitor with the surrounding soil as the

dielectric medium. The sensor measures the charge time, which reflects the soil's dielectric properties, and outputs a scaled frequency (RAW) value. This value is converted to $\varepsilon_\mathrm{eff}$ using a third-order polynomial calibration equation provided in the manual:

$$\varepsilon_\mathrm{eff} = \left(2.887 \times 10^{-9} \times \mathrm{RAW}^3 - 2.080 \times 10^5 \times \mathrm{RAW}^2 + 5.276 \times 10^2 \times \mathrm{RAW} - 43.39\right)^2 \tag{A2}$$

The sensor's reported accuracy is $\pm 1$ unit for $\varepsilon_\mathrm{eff} \in [1, 40]$ and $\pm 15\%$ for values above 40. However, studies have shown

that the TEROS 12 can systematically underestimate dielectric permittivity in highly saturated soils and exhibits increased sensitivity under saline conditions (Cominelli et al., 2024; Fragkos et al., 2024). While the manufacturer lists a nominal sensing volume of approximately $1010\,\mathrm{cm}^3$, experimental evaluations report a smaller effective volume in moist sand (approximately $423\,\mathrm{cm}^3$) and a further reduction in pure water (down to $84\,\mathrm{cm}^3$). The sensor also demonstrates strong thermal stability, with temperature-induced changes in permittivity typically remaining below 1 unit across the 10–40 °C range (Cominelli et al.,

2024). The TEROS 12 also incorporates an internal thermistor embedded in the central needle to measure temperature. These temperature readings range from $-40$ to 60 °C, with an accuracy of $\pm 0.5$ °C from $-40$ to 0 °C and $\pm 0.3$ °C from 0 to 60 °C.

## A3  HydraProbe (Stevens Water)

The HydraProbe (Stevens Water Monitoring Systems) uses coaxial impedance dielectric reflectometry to measure the real and imaginary components of complex dielectric permittivity. It features a coaxial waveguide with four stainless steel tines (0.3 cm

diameter, 5.7 cm length) arranged in a circle around a central tine, protruding from a 4.2 cm metal base plate. A 50 MHz signal





is transmitted through the tines, and the sensor analyzes the amplitude ratio of incident to reflected waves to solve Maxwell's equations. This allows separate estimation of the real ($\varepsilon_r'$) and imaginary ($\varepsilon_r''$) components of $\varepsilon_{\text{eff}}$, computed as follows (von Hippel, 1966; Topp et al., 1980):

$$\varepsilon_{\text{eff}} = \frac{\varepsilon_r'}{2}\left(1 + \sqrt{1 + \left(\frac{\varepsilon_r''}{\varepsilon_r'}\right)^2}\right) \tag{A3}$$

Laboratory tests confirm that the HydraProbe provides precise and consistent permittivity measurements, with inter-sensor variability typically below $\pm0.5$ units and <1% coefficient of variation in fluids (Seyfried and Murdock, 2004). It performs reliably up to soil EC values of $\sim 0.14\,\text{S}\,\text{m}^{-1}$, beyond which accuracy declines. Loss tangent values above $\sim 1.45$ lead to unstable readings. Despite lacking internal temperature correction, temperature effects are minor—e.g., $\sim 0.0077$ units$^\circ\text{C}^{-1}$ in air and up to $\pm0.06\,\text{m}^3\,\text{m}^{-3}$ in saturated clay soils over a $40\,^\circ\text{C}$ range. The nominal sensing volume of the HydraProbe is
approximately $4.0 \times 10^4\,\text{mm}^3$, but it can expand up to $\sim 3.5 \times 10^5\,\text{mm}^3$ depending on soil conditions. The effective sensing volume increases in soils with lower permittivity—such as dry or frozen soils—and contracts in wetter soils with higher permittivity. Temperature is measured via a thermistor in contact with the base plate, with a range of $-40^\circ\text{C}$ to $+75^\circ\text{C}$, an accuracy of $\pm0.3^\circ\text{C}$, and a resolution of $0.1^\circ\text{C}$.

## Appendix B:  Instrumentation Setup

Figure A1 illustrates the insertion depths and orientations of each probe (CS616, HydraProbe, and TEROS 12), along with their standard needle lengths. Accompanying the schematic are actual images of the probes to enhance visual recognition and familiarity with their designs.

## Appendix B:  Derivation of Bai et al.'s Model in Permittivity-Temperature Space

In this section, we present the detailed steps that transform the Eq. (1) from liquid water content-soil temperature space into
permittivity-soil temperature space. The resulting equation expresses the soil's effective permittivity as a function of soil temperature, as shown in Eq.( 4).

### B1   For $T > T_{\text{soil}}$

The simplified form of Eq.( 3) is:

$$\theta_{lw} = A\varepsilon_{\text{eff}}^{\alpha} + B - \theta_{\text{ice}}C \tag{B1}$$

where $A$, $B$, and $C$ are defined as:



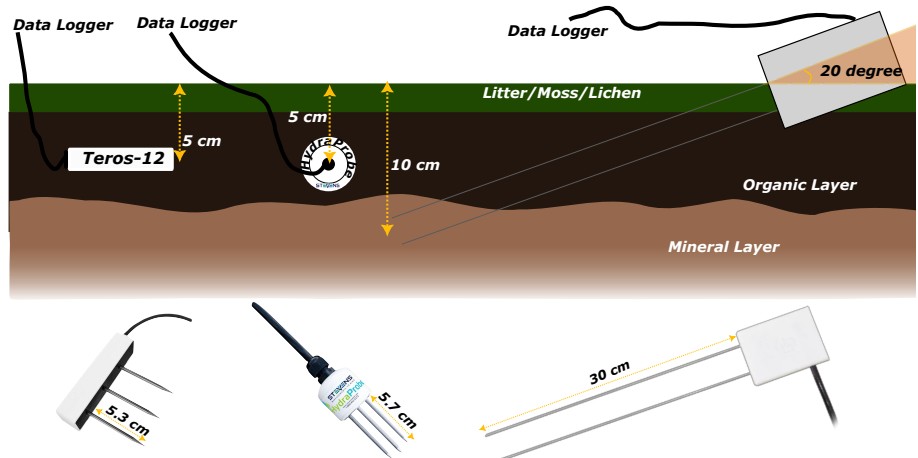

**Figure A1.** Instrumentation setup for soil moisture probes (CS616, HydraProbe, and TEROS 12), with a schematic and corresponding probe images.

$$A = \frac{1}{\varepsilon_{lw}^{\alpha} - \varepsilon_{\text{air}}^{\alpha}}, \tag{B2}$$

$$B = \frac{(n-1)\varepsilon_{\text{soil}}^{\alpha} - n\varepsilon_{\text{air}}^{\alpha}}{\varepsilon_{lw}^{\alpha} - \varepsilon_{\text{air}}^{\alpha}}, \tag{B3}$$

$$C = \frac{\varepsilon_{\text{ice}}^{\alpha} - \varepsilon_{\text{air}}^{\alpha}}{\varepsilon_{lw}^{\alpha} - \varepsilon_{\text{air}}^{\alpha}}. \tag{B4}$$

For $T > T_{\text{soil}}$, we know that $\theta_{\text{ice}} = 0$. Therefore, we have:

$\quad \theta_{\text{lw}} = A\varepsilon_{\text{eff}}^{\alpha} + B \tag{B5}$

Additionally, the liquid water content $\theta_{\text{int}}$ is given by:

$$\theta_{\text{int}} = A\varepsilon_{\text{int}}^{\alpha} + B \tag{B6}$$

Thus, for $T > T_{\text{soil}}$:

$$\varepsilon_{\text{eff}} = \varepsilon_{\text{int}} \tag{B7}$$





## B2   For $T \leq T_{\text{soil}}$

For $T \leq T_{\text{soil}}$, we can express $\theta_{\text{ice}}$ in terms of $\theta_{\text{int}}$ and $\theta_{\text{res}}$:

$$\theta_{\text{ice}} = \theta_{\text{int}} - \theta_{\text{lw}} \tag{B8}$$

**Step 1**: Substitute $\theta_{\text{ice}}$ into $\theta_{\text{res}}$:

$$\theta_{\text{res}} = A\varepsilon_{\text{res}}^{\alpha} + B - (\theta_{\text{int}} - \theta_{\text{res}})C \tag{B9}$$

**Step 2**: Simplify the equation:

$$\theta_{\text{res}} = A\varepsilon_{\text{res}}^{\alpha} + B - \theta_{\text{int}}C + \theta_{\text{res}}C \tag{B10}$$

**Step 3**: Bring like terms together:

$$\theta_{\text{res}}(1 - C) = A\varepsilon_{\text{res}}^{\alpha} + B - \theta_{\text{int}}C \tag{B11}$$

**Step 4**: Solve for $\theta_{\text{res}}$:

$$\theta_{\text{res}} = \frac{A\varepsilon_{\text{res}}^{\alpha} + B - \theta_{\text{int}}C}{1 - C} \tag{B12}$$

**Step 5**: Similarly, for $\theta_{\text{lw}}$:

$$\theta_{\text{lw}} = \frac{A\varepsilon_{\text{eff}}^{\alpha} + B - \theta_{\text{int}}C}{1 - C} \tag{B13}$$

**Step 6**: Substitute these expressions into the model from Bai et al. (2018) for $T \leq T_{\text{soil}}$:

$$\theta_{\text{lw}} = (\theta_{\text{int}} - \theta_{\text{res}})e^{b(T_{\text{soil}} - T_f)} + \theta_{\text{res}} \tag{B14}$$

**Step 7**: Express $\theta_{\text{int}} - \theta_{\text{res}}$:

$$\theta_{\text{int}} - \theta_{\text{res}} = \frac{A\left(\varepsilon_{\text{int}}^{\alpha} - \varepsilon_{\text{res}}^{\alpha}\right)}{1 - C} \tag{B15}$$

**Step 8**: Substitute back and simplify:



$$\frac{A\varepsilon_{\text{eff}}^{\alpha} + B - \theta_{\text{int}}C}{1-C} = \frac{A\left(\varepsilon_{\text{int}}^{\alpha} - \varepsilon_{\text{res}}^{\alpha}\right)}{1-C}e^{b(T_{\text{soil}}-T_f)} + \frac{A\varepsilon_{\text{res}}^{\alpha} + B - \theta_{\text{int}}C}{1-C} \tag{B16}$$

**Step 9**: Multiply both sides by $1-C$ to eliminate denominator:

$$A\varepsilon_{\text{eff}}^{\alpha} + B - \theta_{\text{int}}C = A\left(\varepsilon_{\text{int}}^{\alpha} - \varepsilon_{\text{res}}^{\alpha}\right)e^{b(T_{\text{soil}}-T_f)} + A\varepsilon_{\text{res}}^{\alpha} + B - \theta_{\text{int}}C \tag{B17}$$

**Step 10**: Subtract common terms from both sides:

$$A\varepsilon_{\text{eff}}^{\alpha} = A\left(\varepsilon_{\text{int}}^{\alpha} - \varepsilon_{\text{res}}^{\alpha}\right)e^{b(T_{\text{soil}}-T_f)} + A\varepsilon_{\text{res}}^{\alpha} \tag{B18}$$

**Step 11**: Divide both sides by $A$:

$$\varepsilon_{\text{eff}}^{\alpha} = \left(\varepsilon_{\text{int}}^{\alpha} - \varepsilon_{\text{res}}^{\alpha}\right)e^{b(T_{\text{soil}}-T_f)} + \varepsilon_{\text{res}}^{\alpha} \tag{B19}$$

Thus, we finally arrive at:

$$\varepsilon_{\text{eff}} = \left(\left(\varepsilon_{\text{int}}^{\alpha} - \varepsilon_{\text{res}}^{\alpha}\right)e^{b(T_{\text{soil}}-T_f)} + \varepsilon_{\text{res}}^{\alpha}\right)^{\frac{1}{\alpha}} \tag{B20}$$

*Code and data availability.* Scripts used for analysis and plotting, primarily written in Python 3.9, are available upon request from the authors. The dataset used to create the interactive site map is publicly accessible on GitHub (GitHub Repository) (Salmabadi et al., 2025). The raw sensor outputs, including effective permittivity of bulk soil (computed from the direct raw measurements of the probes), uncalibrated soil moisture data, and soil temperature measurements, along with the final study output (degree of soil freezing), are available upon request for future research applications.

*Author contributions.* H.S.: Conceptualization, Methodology, Software, Data Curation, Formal Analysis, Validation, Visualization, Funding Acquisition, Writing – Original Draft, Writing – Review & Editing. R.P.L.: Methodology, Writing – Review & Editing. A.B.: Supervision, Funding Acquisition, Investigation, Writing – Review & Editing. A.R.: Supervision, Methodology, Funding Acquisition, Investigation, Writing – Review & Editing. A.M.: Investigation, Data Curation, Writing – Review & Editing. C.H.: Investigation, Data Curation, Writing – Review & Editing. B.M.: Investigation, Data Curation, Writing – Review & Editing.

*Competing interests.* The contact author has declared that none of the authors has any competing interests.





*Acknowledgements.* This study was supported by the Natural Sciences and Engineering Research Council of Canada (NSERC) and the Canadian Space Agency (CSA), as well as the Fonds de recherche du Québec – Nature et technologies (FRQNT) through a Ph.D. scholarship
awarded to H.S. We also extend our gratitude to Joshua King, who passed away in February 2023, for his invaluable contributions to this field, particularly for his work on instrumentation in the Trail Valley Creek dataset. We would also like to thank the Centre d'Études Nordiques (CEN) for their logistical support. This work was further supported by contributions from Hydro-Québec, and Environment and Climate Change Canada. Finally, we acknowledge the invaluable assistance of our colleagues Camille Roy, Alex Gélinas, Azza Gorrab, Esteban Hamel Jomphe, and Kayla Wicks whose help was essential during fieldwork.

*Financial support.* This research was supported by the Fonds de recherche du Québec – Nature et technologies (grant no. 330450) and the Canadian Space Agency (grant no. 19FAQCT B19).



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
