# Peer review of "Improving Seasonally Frozen Ground Monitoring Using Soil Freezing Characteristic Curve in Permittivity-Temperature Space"

_EGUsphere, 2025_

## Referee Comment (RC1)

**Review of manuscript "Improving Seasonally Frozen Ground Monitoring Using Soil Freezing Characteristic Curve in Permittivity-Temperature Space" by Salmabadi et al. (submitted to EGUsphere)**

This manuscript describes a comprehensive evaluation of soil moisture and soil temperature measurements at various locations in Canada with the aim of improving the classification of soils according to their freezing state. To this end, the authors consistently reconstructed and uniformly evaluated the freezing characteristic curve at 87 Canadian locations. This enabled them to determine where in Canada the soil is frozen for how long and how long the corresponding freezing and thawing phases last.

Since the fundamental work of R.D. Miller and P.J.Williams & M.W. Smith in the 1970s and 1980s, the freezing processes of soils in relation to the phase change from liquid water to ice and the associated temperature and pressure changes have been relatively well understood. In the 1990s, several doctoral theses were written in Minnesota (E. Spaans), Zurich (D. Stadler), and Uppsala (M. Stähli), where the freezing characteristics of soils were measured—very similar to this work by Salmabadi— and then used in numerical models. Even back then, we had similar discussions to those we are having today: when exactly is soil "frozen"? And how relevant is this partially frozen state? I am very pleased that this topic is being addressed once again in this manuscript.

The methodology for the precise analysis of the freezing state based on the freezing characteristics is described in great detail and is easy to understand. I have hardly any questions or objections to this. Of course, one could perhaps be critical of the assumption in lines 144-145 that the total water content remains constant during the freezing process. We know that this is not the case, but that water is transported from lower soil layers to the freezing front. However, for the methodology used here, I do not think it is a major problem to accept this simplification. More decisive for me are the assumptions in section 2.5.1, a) that hourly values are ultimately aggregated to daily values and thus only determined on a daily basis whether the soil is frozen, partially frozen or unfrozen, and b) that the threshold values are set at $p=0.1$ and $0.9$ respectively. This makes sense to me, but I would still be curious to know whether there is a sensitivity analysis for these assumptions and threshold values.

Now to the results. The various freezing curves measured at a total of 87 locations are pooled into four large regions. This is then used to make statements about how these large regions differ in terms of freezing and thawing. I find it relatively bold to make such broad regional statements based on such a small number of sensors, which also represent a very local scale and are unevenly distributed. Is it really justified to say that in the eastern boreal forest, soils typically freeze within a very small temperature range, while in the western boreal forest, freezing is more gradual?

This measured freezing behavior, which varies from location to location and from large region to large region, naturally has various causes, as explained in the discussion on line 365. It has a lot to do with soil properties, but also with the history (antecedent moisture content of the soil). However, when it comes to the question of "How many days per year is the soil frozen, partially frozen, or unfrozen?", two important factors come into play that are hardly discussed in the text: snow cover and air temperature. This aspect could be emphasized a little more in the manuscript. It would certainly be useful for the reader to learn more about the meteorological conditions and snow cover at the various measurement sites. Ultimately, the freezing curves are also influenced by these factors.

Overall, I really like the study. The investigation of the freezing characteristics at the various Canadian locations is very thorough and undoubtedly adds value. As far as I can tell, the manuscript is linguistically flawless and well illustrated. Thank you very much.

Minor issues:

Line 130: "are are available" ("are" twice, remove one of them)

---

## Author Comment (AC1)

**In Situ Monitoring of Seasonally Frozen Ground Using Soil Freezing Characteristic Curve in Permittivity-Temperature Space**

**Authors' Response to Reviewer 1**

**Reviewer #1's comment**

This manuscript describes a comprehensive evaluation of soil moisture and soil temperature measurements at various locations in Canada with the aim of improving the classification of soils according to their freezing state. To this end, the authors consistently reconstructed and uniformly evaluated the freezing characteristic curve at 87 Canadian locations. This enabled them to determine where in Canada the soil is frozen for how long and how long the corresponding freezing and thawing phases last. Since the fundamental work of R.D. Miller and P.J. Williams & M.W. Smith in the 1970s and 1980s, the freezing processes of soils in relation to the phase change from liquid water to ice and the associated temperature and pressure changes have been relatively well understood. In the 1990s, several doctoral theses were written in Minnesota (E. Spaans), Zurich (D. Stadler), and Uppsala (M. Stähli), where the freezing characteristics of soils were measured—very similar to this work by Salmabadi—and then used in numerical models. Even back then, we had similar discussions to those we are having today: when exactly is soil "frozen"? And how relevant is this partially frozen state? I am very pleased that this topic is being addressed once again in this manuscript. Overall, I really like the study. The investigation of the freezing characteristics at the various Canadian locations is very thorough and undoubtedly adds value. As far as I can tell, the manuscript is linguistically flawless and well illustrated. Thank you very much.

**Response:**

We sincerely thank the reviewer for these kind words and for recognizing the value of revisiting this important topic. We appreciate the positive assessment of our work and the thorough evaluation provided.

**Reviewer #1's comment**

The methodology for the precise analysis of the freezing state based on the freezing characteristics is described in great detail and is easy to understand. I have hardly any questions or objections to this. Of course, one could perhaps be critical of the assumption in lines 144–145 that the total water content remains constant during the freezing process. We know that this is not the case, but that water is transported from lower soil layers to the freezing front. However, for the methodology used here, I do not think it is a major problem to accept this simplification. More decisive for me are the assumptions in section 2.5.1, a) that hourly values are ultimately aggregated to daily values and thus only determined on a daily basis whether the soil is frozen, partially frozen or unfrozen, and b) that the threshold values are set at p = 0.1 and 0.9 respectively. This makes sense to me, but I would still be curious to know whether there is a sensitivity analysis for these assumptions and threshold values.

**Response:**

We thank the reviewer for acknowledging that the assumption of constant total water content is an acceptable simplification for our methodology. We agree that water migration toward the freezing front occurs in reality; however, as the reviewer notes, this does not significantly affect the validity of our approach for characterizing the freezing dynamics within the monitored soil layer. We have clarified this assumption in the *Data Preprocessing* section and explicitly acknowledged it as a limitation in the *Discussion*.

Regarding temporal resolution, we confirm that all analyses were performed using *hourly* data, and the same temporal resolution is maintained in the final dataset that will be made publicly available. Aggregation to daily values was done solely for visualization and summarization purposes, providing a clearer overview of freezing patterns across networks.

Concerning threshold sensitivity, we agree that the initial thresholds (p = 0.1 for unfrozen and p = 0.9 for frozen) were somewhat arbitrary. We therefore conducted a sensitivity

analysis to assess their influence. Based on the shape of the fitted SFCCs, we tested alternative thresholds of  $p = 10^{-6}$  for unfrozen (representing a near-zero probability of freezing) and p = 0.75 for frozen. The aggregated daily results showed negligible changes when using p = 0.75 instead of p = 0.9, confirming that our classification and derived statistics are robust within reasonable threshold ranges.

**Reviewer #1's comment**

Now to the results. The various freezing curves measured at a total of 87 locations are pooled into four large regions. This is then used to make statements about how these large regions differ in terms of freezing and thawing. I find it relatively bold to make such broad regional statements based on such a small number of sensors, which also represent a very local scale and are unevenly distributed. Is it really justified to say that in the eastern boreal forest, soils typically freeze within a very small temperature range, while in the western boreal forest, freezing is more gradual?

**Response:**

We agree with this concern and thank the reviewer for highlighting it. In the original manuscript, we intended to caution readers against overgeneralizing our results across entire landscapes or biomes. However, as the reviewer correctly observed, presenting our findings by ecozone/land-cover type inadvertently conveyed a broader regional interpretation than warranted by the spatial density of our observations.

To address this, we have thoroughly revised the manuscript to avoid overgeneralization and ensure that results are described strictly at the network level.

**Changes made:**

• Restructured results presentation: Results are now reported on a *network-by-network* basis rather than by ecozone/land-cover group. For instance, Table 2 has been updated to summarize curve fitting performance metrics for each network, and Figures 4 and 5 have been revised accordingly to reflect network-specific results.

**Modified table in manuscript (Results):**

| Land Cover TypeNetwork                              | $R^2$                                       | RMSE                               | MAE                                           |
|-----------------------------------------------------|---------------------------------------------|------------------------------------|-----------------------------------------------|
| Eastern boreal forest BJ                            | $\underbrace{0.85}_{0.93}\underbrace{0.93}$ | <del>1.25 (18.28</del> 1.20 (7.2%) | <del>0.81 (11.85</del> 0.94 (5.7%)            |
| Western boreal forest $\underbrace{\mathrm{BT}}_{}$ | $\underbrace{0.94}_{}\underbrace{0.95}_{}$  | $0.29 \ (8.033.7\%)$               | $0.18 \ (4.992.3\%)$                          |
| Prairies_CP                                         | $\underbrace{0.82}_{}$                      | 0.80 (8.6%)                        | $\underbrace{0.61}_{}\underbrace{(6.6\%)}_{}$ |
| $\widecheck{\mathrm{FM}}$                           | $\underbrace{0.66}_{}$                      | 2.34 (11.7%)                       | 1.70 (8.5%)                                   |
| $\widetilde{\operatorname{GR}}$                     | $\overset{0.67}{\sim}\!\!\!\!\sim$          | 2.13 (8.4%)                        | 1.17 (4.6%)                                   |
| KN                                           | 0.95                                        | <del>1.07 (10.94</del> 1.14 (3.8%) | 0.65 (6.65 0.71 (2.4%)                        |
| Tundra LR                                           | $\underbrace{0.87}_{\sim}$                  | 0.43 (7.4%)                        | 0.25(4.2%)                                    |
| TV                                                  | 0.86                                        | <del>1.71 (39.20</del> 2.07 (6.5%) | 0.87 (19.941.19 (3.8%)                        |
| Overall                                             | 0.95                                        | <del>1.16 (14.58</del> 1.26 (4.0%) | 0.64 (8.050.72(2.3%)                          |
| Overall (balanced)                                  | $\underbrace{0.89}_{\sim}$                  | 1.54 (4.9%)                        | 0.86(2.7%)                                    |

**Modified figures in manuscript (Results):**

• Revised terminology throughout: The Results section now explicitly refers to individual monitoring networks (e.g., "James Bay (BJ) network," "Candle Lake (BT) network") instead of broader regions such as "eastern boreal forest" or "western

boreal forest." For example, the previous statement "eastern boreal forests freeze gradually" has been replaced by: "The James Bay (BJ), Montmorency Forest (FM), Chapleau (CP), and La Romaine (LR) networks, located within the eastern boreal forest ecozone, exhibit...," clarifying that our findings pertain to the specific monitored locations rather than the entire region.

**Modified text in manuscript (Results):**

To illustrate the application of our SFCC model (Eq. 4) and its integration with in situ data, we presented five present four example sites from different monitoring networks, each exhibiting distinct freezing behaviors: eastern boreal forest (networks and ecozones, each representing a distinct freezing regime: EC17 from KN (prairie; Fig. Fig. 7), 6), BT17 from BT (western boreal forest; Fig. 7), prairie (BJ01 from BJ (eastern boreal forest; Fig. 98), and tundra (GR01 from GR (tundra; Fig. 9). Each example consisted of includes two panels: the first panelpanel (a) depicted shows the fitted SFCC overlaid on the processed in situ measurements of in situ soil temperature and permittivity, with vertical lines indicating  $T_f$  and  $T_{\rm res}$ . The second panel  $\epsilon_{\rm eff}$  measurements, with Tf marked; panel (b) displayed the presents the corresponding time series of soil temperature for the same site, color-coded by the probability of frozen ground (degree of soil freezing ). To further summarize the results, we included, freezing probability, ERA5 air temperature, and IMS snow cover. To summarize all networks, Fig. 10, which shows the monthly average of the probability of frozen ground over the entire data period for all sites within each network. This visualization provided insight into the average freezing and thawing patterns at each station and network. In the eastern boreal forest, soils rarely freeze completely during winter (Fig. 10), reflecting high water retention and the insulating effect of snow and vegetation cover. This leads to either prolonged transitional states displays the monthly mean freezing probability, with monthly mean air and soil temperatures overlaid on each tile. To compare networks quantitatively, soil states were classified at hourly resolution as frozen if  $P_{\text{frozen}} > 0.75$ , unfrozen if  $P_{\text{frozen}} < 10^{-6}$ , and transitional (partially frozen) otherwise. Daily states were assigned by majority rule.

BJ, CP, FM, and LR—a phenomenon known as the zero curtain effect located in the eastern boreal forest—or unfrozen soil throughout the year. On average, we recorded 23 frozen days and 46 transitional days in this region remained predominantly unfrozen or partially frozen throughout the freezing season, with no periods of complete freezing despite persistent snow cover (> 90% from December to February) and subzero air temperatures. Their soils stayed near  $0^{\circ}$ C, yielding moderate freezing probabilities ( $\leq 0.65$ ). For instance, FM and CP remained partially frozen for approximately 125 days, while BJ and LR showed shorter transitional periods (≈ 100 days) with  $\approx 75$  unfrozen days. In contrast, the western borealforest, characterized by drier conditions compared to its eastern counterpart, experiences more extensive freezing while still retaining some transitional states. On average, we recorded 73 frozen days and 76 transitional daysat these sites. As expected, tundra sites exhibited the longest frozen periods, with an average of 145 frozen days due to consistently low temperatures, along with 52 transitional days. Prairie soils began freezing earlier than boreal forest soils, likely due to the absence or shallow depth of snow cover and the lack of vegetation, as these sites are primarily agricultural lands. This lack of insulation made prairiesoils more susceptible to air temperature fluctuations (Fig 6), allowing soil temperatures to drop more rapidlyBT (western boreal) exhibited extensive freezing from December onward (90 frozen and 30 transitional days), coinciding with subzero air temperatures and complete snow cover. Tundra networks (TV, GR) recorded the coldest soil and air temperatures from December to February (soil and air consistently below -2 °C and -10 °C, respectively) and persistent snow cover ( $\approx 100\%$ ), resulting in almost continuous frozen conditions (> 0.95) lasting 135 and 95 days for TV and GR respectively. The KN (prairie) network showed an intermediate response: although mean air temperatures ( $\approx -3^{\circ}$ C) were comparable to eastern forest networks, soil cooling began earlier (November,  $T_{\rm Soil} \approx 0.9^{\circ}$ C) and remained frozen through February (> 0.9) under shallow or intermittent snow (50 – 95%). On average, we recorded 71 frozen days and 71 transitional days in the prairies. Additionally, prairie soils thaw earlier than other landcover types, reflecting their sensitivity to air temperature variations about 70 days were classified as frozen and 60 days as transitional at KN. Overall, while all networks experienced similar subzero air temperatures and persistent snow cover from December to February, only tundra (TV, GR), western boreal (BT), and prairie (KN) networks exhibited sustained frozen states, whereas eastern forest networks (BJ, CP, LR, FM) remained largely in transitional or unfrozen conditions.

These revisions ensure that all results are contextualized appropriately to the spatial scale of the dataset and that no unintended regional generalizations remain in the text.

**Reviewer #1's comment**

This measured freezing behavior, which varies from location to location and from large region to large region, naturally has various causes, as explained in the discussion on line 365. It has a lot to do with soil properties, but also with the history (antecedent moisture content of the soil). However, when it comes to the question of "How many days per year is the soil frozen, partially frozen, or unfrozen?", two important factors come into play that are hardly discussed in the text: snow cover and air temperature. This aspect could be emphasized a little more in the manuscript. It would certainly be useful for the reader to learn more about the meteorological conditions and snow cover at the various measurement sites. Ultimately, the freezing curves are also influenced by these factors.

**Response:**

We greatly appreciate this valuable feedback. The reviewer is correct that snow cover and air temperature are among the most important environmental controls on soil freezing dynamics. These factors were underrepresented in the original version, and we have made substantial revisions to strengthen their treatment in the manuscript.

**Changes made:**

• Expanded Introduction: A new paragraph was added in the *Introduction* to expand the literature review and explicitly highlight the roles of air temperature, snow cover, and soil moisture as key controls on ground thermal regimes and soil freezing behavior.

**Modified text in manuscript (Introduction):**

Beyond air temperature, which governs convective heat loss, soil freezing is primarily regulated by ground surface cover including snowpack, vegetation canopy and litter, moss, and organic (humus) layers as well as by soil moisture (MacKinney, 1929). Collectively, ground surface cover act as a thermal buffer, moderating soil–atmosphere heat exchange, conserving water, and reducing frost penetration (Fu et al., 2018). The insulating influence of snow, vegetation, litter, moss, and organic layers on soil temperature and moisture retention is well documented (Zhang, 2005; Decker et al., 2003; Flerchinger and Pierson, 1991; MacKinney, 1929; Gornall et al., 2007; Park et al., 2018; Lawrence et al., 2008; Oogathoo et al., 2022). Meanwhile, soil moisture exerts a dual control: its latent heat delays freezing onset, whereas ice formation increases thermal conductivity and accelerates subsequent cooling (Kersten, 1949; Lei et al., 2020).

- Added ancillary datasets: Two new datasets were integrated to provide environmental context for each network:
  - IMS (Interactive Multisensor Snow and Ice Mapping System) daily snow cover at 4 km resolution.
  - ERA5-Land hourly 2-meter air temperature at 0.25° resolution.

These datasets were merged with the hourly in situ measurements to provide comprehensive meteorological information.

**Modified text in manuscript (Methodology):**

Air temperature data were obtained from the ERA5-Land reanalysis developed by the European Centre for Medium-Range Weather Forecasts (ECMWF; (C3S, 2018)). ERA5-Land provides hourly 2 m air temperature at  $0.25^{\circ}$  spatial resolution and assimilates global observations within a physics-based numerical model to produce a consistent reanalysis extending from 1940 to the present. Snow cover data were derived from the Interactive Multisensor Snow and Ice Mapping System (IMS) produced by the U.S. National Ice Center (U.S. National Ice Center, 2004), which provides daily binary snow-cover maps for the Northern Hemisphere at 4 km resolution since 2004. The IMS product integrates multisensor satellite imagery and in situ observations. For each study site, snow-cover and air-temperature values were extracted from the corresponding IMS and ERA5-Land grid cells.

• Enhanced figures: Figures 6–9 were revised to include time series of ERA5-Land air temperature alongside soil temperature, and to indicate periods of snow cover derived from IMS data.

**Modified figure in manuscript (Results):**

• Revised network-level heatmap (Figure 10): The network summary heatmap now displays monthly mean soil temperature, air temperature, and snow cover percentage, providing an integrated view of these controlling factors.

Modified figure in manuscript (Results):

- Integrated into Results: The section *Model Application to Field Data* now explicitly reports air temperature conditions, snow cover duration, and their relationships to the observed freezing probabilities for each network.
- Expanded Discussion: A new paragraph was added discussing how the combined effects of air temperature, snow cover, organic layer thickness, and soil moisture influence the freezing dynamics across our networks. This synthesis strengthens our interpretation of spatial differences in freezing behavior.

**Modified text in manuscript (Discussion):**

Differences in soil freezing across the networks primarily reflect the combined effects of insulation and moisture rather than air temperature alone. Eastern boreal networks (BJ, FM, LR, and CP) remained largely unfrozen throughout winter due to thicker moss and organic layers, denser canopy cover, and higher soil moisture. These features collectively buffer ground heat loss, dampen temperature fluctuations, prolong the zero curtain phase, and keep the soil in a transitional state for extended periods. This interpretation aligns with modeling results from boreal forests in eastern Canada, where soils were shown to remain near 0°C throughout winter despite mean air temperatures around -16°C (Oogathoo et al., 2022; Lawrence et al., 2008). Such multilayer insulation reduces conductive and radiative heat exchange between the atmosphere and the soil, thereby limiting frost penetration even under severe cold conditions. In contrast, BT—a dry boreal network with sparse vegetation and thin organic horizons—and KN, which lacks vegetation and organic cover, exhibited earlier and deeper freezing. Tundra networks (GR and TV) experienced prolonged freezing driven by extreme cold and minimal insulation; soil freezing began almost immediately after air temperatures fell below 0°C, although GR's higher soil moisture delayed freeze onset relative to TV.

**References**

- C3S. ERA5 hourly data on single levels from 1940 to present, 2018. URL https://cds.climate.copernicus.eu/doi/10.24381/cds.adbb2d47.
- K. L. M. Decker, D. Wang, C. Waite, and T. Scherbatskoy. Snow Removal and Ambient Air Temperature Effects on Forest Soil Temperatures in Northern Vermont. Soil Science Society of America Journal, 67(4):1234–1242, July 2003. ISSN 03615995. doi: 10.2136/sssaj2003.1234. URL http://doi.wiley.com/10.2136/sssaj2003.1234.

- G.N. Flerchinger and F.B. Pierson. Modeling plant canopy effects on variability of soil temperature and water. *Agricultural and Forest Meteorology*, 56(3-4):227-246, September 1991. ISSN 01681923. doi: 10.1016/0168-1923(91)90093-6. URL https://linkinghub.elsevier.com/retrieve/pii/0168192391900936.
- Qiang Fu, Renjie Hou, Tianxiao Li, Ruiqi Jiang, Peiru Yan, Ziao Ma, and Zhaoqiang Zhou. Effects of soil water and heat relationship under various snow cover during freezing-thawing periods in Songnen Plain, China. *Scientific Reports*, 8(1):1325, January 2018. ISSN 2045-2322. doi: 10.1038/s41598-018-19467-y. URL https://www.nature.com/articles/s41598-018-19467-y.
- J. L. Gornall, I. S. Jónsdóttir, S. J. Woodin, and R. Van Der Wal. Arctic mosses govern below-ground environment and ecosystem processes. *Oecologia*, 153(4):931–941, October 2007. ISSN 0029-8549, 1432-1939. doi: 10.1007/s00442-007-0785-0. URL http://link.springer.com/10.1007/s00442-007-0785-0.
- Miles S. Kersten. Thermal Properties of Soils, June 1949. URL https://hdl.handle.net/11299/124271.
- David M. Lawrence, Andrew G. Slater, Vladimir E. Romanovsky, and Dmitry J. Nicolsky. Sensitivity of a model projection of near-surface permafrost degradation to soil column depth and representation of soil organic matter. *Journal of Geophysical Research: Earth Surface*, 113(F2):2007JF000883, June 2008. ISSN 0148-0227. doi: 10.1029/2007JF000883. URL https://agupubs.onlinelibrary.wiley.com/doi/10.1029/2007JF000883.
- Dawei Lei, Yugui Yang, Chengzheng Cai, Yong Chen, and Songhe Wang. The Modelling of Freezing Process in Saturated Soil Based on the Thermal-Hydro-Mechanical Multi-Physics Field Coupling Theory. *Water*, 12(10):2684, September 2020. ISSN 2073-4441. doi: 10.3390/w12102684. URL https://www.mdpi.com/2073-4441/12/10/2684.
- A. L. MacKinney. Effects of Forest Litter on Soil Temperature and Soil Freezing in Autumn and Winter. *Ecology*, 10(3):312–321, July 1929. ISSN 0012-9658, 1939-9170. doi: 10.2307/1929507. URL https://esajournals.onlinelibrary.wiley.com/doi/10.2307/1929507.

- Shalini Oogathoo, Daniel Houle, Louis Duchesne, and Daniel Kneeshaw. Evaluation of simulated soil moisture and temperature for a Canadian boreal forest. *Agricultural and Forest Meteorology*, 323:109078, August 2022. ISSN 01681923. doi: 10.1016/j.agrformet.2022.109078. URL https://linkinghub.elsevier.com/retrieve/pii/S0168192322002660.
- Hotaek Park, Samuli Launiainen, Pavel Y. Konstantinov, Yoshihiro Iijima, and Alexander N. Fedorov. Modeling the Effect of Moss Cover on Soil Temperature and Carbon Fluxes at a Tundra Site in Northeastern Siberia. *Journal of Geophysical Research: Biogeosciences*, 123(9):3028–3044, September 2018. ISSN 2169-8953, 2169-8961. doi: 10.1029/2018JG004491. URL https://agupubs.onlinelibrary.wiley.com/doi/10.1029/2018JG004491.
- U.S. National Ice Center. IMS Daily Northern Hemisphere Snow and Ice Analysis at 1 km, 4 km, and 24 km Resolutions, Version 1, 2004. URL https://nsidc.org/data/G02156/versions/1.
- Tingjun Zhang. Influence of the seasonal snow cover on the ground thermal regime: An overview. Reviews of Geophysics, 43(4):2004RG000157, December 2005. ISSN 8755-1209, 1944-9208. doi: 10.1029/2004RG000157. URL https://agupubs.onlinelibrary.wiley.com/doi/10.1029/2004RG000157.

---

## Author Comment (AC2)

**In Situ Monitoring of Seasonally Frozen Ground Using Soil Freezing Characteristic Curve in Permittivity-Temperature Space**

**Authors' Response to Reviewer 2**

**Reviewer #2's comment**

I must confess I've had difficulties understanding the goal of the work presented in the manuscript. After having read what is presented as "advancement [that] addresses a critical gap in cold regions science", I still do not know what new things I have learned, and what is the purported breakthrough. As the remote sensing community appears to be one of the target users of this innovation, what exactly should they do differently now, and how?

**Response:**

We appreciate the reviewer's observation and agree that the original version did not clearly convey the study's goal and contribution. We have substantially revised and restructured the manuscript to clarify our objectives and avoid overstating the novelty. Our goal is now explicitly stated: we applied the Soil Freezing Characteristic Curve (SFCC) directly in permittivity—temperature space to enable continuous, in-situ monitoring of seasonally frozen ground using dielectric sensors, without the need for calibration to estimate liquid or ice water content. This approach allows direct use of field measurements in terms of dielectric permittivity, which is already the primary observable in most soil moisture probes. While SFCC models are well established, their long-term, in-situ application across diverse Canadian ecozones remains scarce. Our study therefore contributes a multi-year, multi-network dataset (87 sites across 8 monitoring networks) that provides hourly probabilities of soil freezing and associated continuous soil-state classifications.

The revised manuscript clarifies that our contribution does not introduce new physical concepts of soil freezing, but rather applies the well-established SFCC to field observations to monitor soil states in situ. The resulting dataset provides high-temporal-resolution ground-truth data that can directly support the remote sensing community in evaluating and training freeze—thaw retrieval algorithms. Current validation practices typically rely on air temperature (often from land surface models) or soil temperature measurements

alone (see the next comment for further details), which overlook the transitional, partially frozen states that persist across all our networks—particularly within the eastern boreal forests of Canada.

**Specific revisions made:**

• The title was changed to accurately reflect the study's focus without overstating its novelty.

**Modified text in manuscript (Title):**

Improving In Situ Monitoring of Seasonally Frozen Ground Monitoring Using Soil Freezing Characteristic Curve in Permittivity-Temperature Space

• The Abstract, Introduction, Results, and Conclusion sections were rewritten and reorganized to more clearly articulate our objectives, workflow, and findings, and to better support the revised narrative. For brevity, we include the revised Abstract below as an example, while the corresponding sections (Introduction, and Results) have been fully revised in the manuscript and can be consulted there for detailed changes.

**Modified text in manuscript (Abstract):**

Frozen ground, a key indicator of climate change, profoundly influences ecological, hydrological, and carbon flux processes in cold regions. However, traditional monitoring methods, which rely on a binary 0°C soil temperature threshold, fail to capture the complexities of soil freezing, such as freezing point depression and transitional states where water and ice coexist. This study introduces a framework that fits a theoretical Soil Freezing Characteristic Curve (SFCC ) in permittivity temperature space to siteand cycle-specific The Soil Freezing Characteristic Curve (SFCC), which defines the relationship between liquid water content and subzero temperatures, provides a framework for understanding soil freezing processes. However, accurately measuring liquid water content in frozen soils under field conditions remains challenging. We therefore recast an empirical SFCC model into permittivity-temperature space and fitted it to in situ measurements. This approach enables the quantification of the degree of soil freezing and the classification of soil states as frozen, unfrozen, or in transition (partially frozen). We analyzed 135 freezing cycles from from eight monitoring networks (87 sites, each equipped with permittivity-based soil moisture probes. These sites are part of eight monitoring networks spanning diverse Canadian landscapes, including eastern boreal forests (Montmorency Forest, La Romaine, James Bay, Chapleau), western boreal forests (Candle Lake), prairies (Kenastonsites) spanning Canadian boreal forests, prairies, and tundra ecozones, encompassing 96 freezing cycles measured with three sensor types (HydraProbe, TEROS12, and CS616). Using Bayesian hierarchical partial pooling, we derived stabilized estimates of key SFCC parameters: the freezing onset temperature  $(T_f)$  and the shape factor (b), which controls transition sharpness. Network-level  $T_f$  ranged from 0.15 to 0.44°C, while b varied from 0.92 to 3.47°C-1, reflecting distinct freezing regimes across ecozones. During the six-month freezing season (1 September-1 March), the James Bay (BJ), Montmorency Forest (FM), Chapleau (CP), and tundra regions (Trail Valley Creek

and George River). On average, eastern boreal forest sites exhibited prolonged unfrozen and transitional states due to high soil moistureretention and insulation from snow and vegetation cover (23 frozen days, 46 transitional days) La Romaine (LR) networks, located in eastern boreal forests with thick organic layers and high moisture, remained predominantly unfrozen (70 days) or in transitional states (110 days) throughout winter despite persistent snow cover (>90%) and subzero air temperatures. In contrast, western boreal forest sites experienced more extensive freezing under drier conditions (73 frozen days, 76 transitional days) Prairie sites displayed equal durations of frozen and transitional states (71 days each), while tundrasites had the longest frozen periods (145 frozen days, 52 transitional days). Notably, transitional periods lasted as long as or even longer than frozen ones, underscoring the limitations of binary classifications. Furthermore, the traditional 0°C threshold misclassified transitional soil states, overestimating frozen days by over 87% in prairie and western boreal regions, and unfrozen days by 86% in the eastern boreal forest. In tundra, the bias was more balanced, with 64% and 36% of transitional daysmisclassified as unfrozen and frozen, respectively. This SFCC-based framework enhances seasonally frozen ground monitoring, offering deeper insights into soil freeze-thaw dynamics. These advancements have implications for improving elimate change assessments, refining carbon flux models, the Trail Valley Creek (TV) and training and validating remote sensing products. Additionally, the resulting database of soil states from this study provides a valuable resource for advancing frozen ground research, particularly in remote sensing and ecosystem modeling efforts. George River (GR) networks, located in tundra, exhibited prolonged frozen conditions (115 days) under extreme cold, though GR's higher moisture delayed freezing relative to TV under similar air temperature conditions. The Candle Lake (BT) network, located in western boreal forest, and the Kenaston (KN) network, located in prairies, showed intermediate responses, with BT experiencing 90 frozen and 30 transitional days, and KN averaging 70 frozen and 60 transitional days despite comparable air temperatures to the eastern boreal networks. These contrasting patterns reflect the combined effects of insulation layer such as snowpack, vegetation canopy, litter, and organic layers, together with moisture rather than air temperature alone, demonstrating how ground surface properties modulate soil thermal regimes. This framework provides a reproducible field-based approach to quantify seasonal surface soil freezing processes and a dataset for model and remote sensing evaluation.

• The *Conclusions* section was simplified to directly state how this study supports remote-sensing applications rather than presenting it as a conceptual breakthrough.

**Modified text in manuscript (Conclusions):**

This study applied an SFCC in permittivity-temperature space to enable robust monitoring of frozen ground states using standard dielectric sensors, transitional, and unfrozen days. Such a perspective provides a dynamic and accurate representation of soil without the calibration challenges inherent in estimating liquid water content. The key insight from our multi-network analysis is that variations in freezing behavior are dominated by local ground surface properties rather than regional air temperature patterns. Importantly, the transitional (partially frozen) state accounts for the majority of the freezing season in eastern boreal networks and persists for at least one month even in western boreal and tundra networks—dynamics that binary frozen/unfrozen classifications fail to capture. These findings reinforce that air temperature alone cannot predict frozen ground extent, demonstrating that remote sensing products and land surface models must account for spatial variations in ground surface properties to accurately represent freeze-thaw behavior, acknowledging the frequent transitions and short-term fluctuations that occur in natural settings. This finer-scale approach more closely aligns with the actual processes

taking place in the soil, allowing for improved monitoring and understanding of seasonally frozen ground conditions.

This study introduces a new framework for seasonally frozen ground monitoring that moves beyond the traditional binary classification approach, offering a more nuanced understanding of soil freeze-thaw-dynamics at regional scales. The practical value of this approach lies in its compatibility with widely deployed sensor networks and its systematic, straightforward methodology for constructing SFCCs from in situ measurements. Numerous soil monitoring networks across cold regions (e.g., RISMA, SNOTEL, AmeriFlux) already measure both soil temperature and dielectric permittivity. These existing infrastructures could readily adopt the methodology presented in this study to monitor seasonally frozen ground. This is particularly important given the rapid warming of high-latitude regions and the need for ground-truth evaluation of satellite-based freeze-thaw products, which currently rely primarily on air or soil temperature observations for training and evaluation (Rautiainen et al., 2025; Donahue et al., 2023; Roy et al., 2020; Kou et al., 2017; Gao et a . By integrating in situ soil temperature and permittivity measurements through an SFCC model, we can now quantify the degree of soil freezing and identify crucial transitional states - a capability particularly valuable for understanding shoulder season processes and carbon flux dynamics. This advancement addresses a critical gap in cold regions science, where accurate characterization of soil states directly impacts our understanding of hydrological processes, ecosystem responses, and carbon cycling. The framework's ability to detect and quantify transitional states, which we found can persist as long as fully frozen periods, has significant implications for improving climate models, particularly their representation of shoulder season biogeochemical processes. Additionally, this methodology provides a robust foundation for validating and improving remote sensing products, potentially enabling more accurate regional and global assessments of frozen ground conditions. Looking forward, this approach opens new avenues for integrating ground-based observations with satellite data, ultimately advancing our ability to monitor and predict cold region responses to climate change.

We believe these changes now make the purpose, scope, and value of the study transparent to both the cold-regions and remote-sensing communities.

**Reviewer #2's comment**

As far as I understood, one of the reported 'novelties' is that the soil moisture doesn't switch from fully unfrozen to fully frozen at 0 °C. This is elementary knowledge in physics and in permafrost science. Additionally, I am presented with evidence that dielectric permittivity, which relates to unfrozen soil moisture fraction, changes gradually over a range of temperatures. Once again, what is the novelty of this?

**Response:**

We agree with the reviewer that the gradual nature of soil freezing and the existence of partially frozen states are well-established in cold-region science and permafrost research. These concepts are not the novelty of our study. Our intent in the original manuscript was to highlight a gap between established cold-region science and current remote-sensing evaluation practices. Many remote-sensing freeze-thaw studies—from early work (Kim et al., 2011; Zhang and Armstrong, 2001) to recent applications (Taghipourjavi et al., 2024; Gao et al., 2020; Kou et al., 2017; Roy et al., 2020; Derksen et al., 2017)—rely on 0°C soil or air temperature thresholds for training and evaluation, without accounting for partially frozen states. Only recently have researchers begun integrating soil moisture, temperature through SFCCs into freeze-thaw model evaluation (Rautiainen et al., 2025). Our study demonstrates that transitional (partially frozen) states are comparably significant to fully frozen and unfrozen states across our networks; indeed, our eastern boreal networks never reached complete freezing during our observation period. However, we acknowledge that emphasizing these well-known physical principles in the Introduction distracted from our actual contribution.

To address this, we made the following revisions:

• **Deleted** the paragraph in the *Introduction* that overstated the gradual-freezing concept. Based on the revised narrative of our study, this paragraph no longer aligns with our objectives, and removing it helps focus readers on our actual contribution.

**Modified text in manuscript:**

Traditionally, the state of the soil has been defined through a single measurement of soil temperature within the top few centimeters of the ground (Andersland and Ladanyi, 2003). Soil is labeled as frozen if the soil (or air temperature) is below 0°C and unfrozen if above. This approach is widely used in numerous studies, particularly in remote sensing, to monitor seasonally frozen ground conditions (Kim et al., 2011; Taghipourjavi et al., 2024; Zhang and Armstrong, 2001; Gao et al., 2020; Kou et al., 2017). However, the soil freezing and thawing process is not binary, and using a single threshold of 0°C—the freezing point of pure water—is not sufficiently accurate (Pardo Lara et al., 2020; Mavrovic et al., 2020). In natural environments, soil typically begins to freeze at temperatures below 0°C (Dobiński, 2020), a phenomenon known as soil freezing point depression, which occurs due to adsorption, capillary action, adhesive and cohesive forces, and osmotic effects (Tian et al., 2014; Bouyoucos and McCool, 1915). Moreover, freezing occurs over a range of temperatures due to the presence of various types of water in the soil—hygroscopic (unfreezable) water, capillary water, water in soil pores, and gravitational water—all of which behave differently during freezing. This diversity creates a transitional zone where water and ice coexist, challenging the traditional binary classification (Tian et al., 2014; Zhang et al., 2019; Bouyoucos and McCool, 1915).

Considering soil freezing point depression, rather than 0°C threshold, would enhance the accuracy of detecting the soil's state, thereby minimizing false positives and negatives in assessments. Additionally, detecting the transitional period is crucial for accurately determining "zero curtain" periods. The zero curtain, observed during shoulder seasons, is a phase when soil temperatures hover around the soil's freezing point regardless of air temperatures. This period is critical as the soil's near-freezing temperature sustains microbial activity (Schimel and Mikan, 2005) ,significantly impacting carbon dioxide fluxes (Arndt et al., 2023). Studies show that carbon emissions during the zero curtain in the fall can match or exceed those of the rest of winter (Arndt et al., 2023; Mavrovic et al., 2023). Furthermore, an observed increase in the duration of the zero curtain period extending into the fall and winter seasons leads to higher carbon emissions during the non-growing season, underscoring the importance of accurately identifying these periods (Arndt et al., 2023).

- Removed all language throughout the manuscript that could be interpreted as claiming discovery of partial freezing behavior.
- Reframed our contribution as providing a practical and systematic framework for constructing SFCCs from in situ dielectric measurements across diverse environmental settings, rather than introducing a new physical concept.
- Expanded the discussion of environmental factors (e.g., organic matter and snow cover) regulating freezing behavior, which are directly relevant to our study's focus on field-based monitoring.

These changes ensure that the revised manuscript correctly reflects that the novelty lies in the operational implementation of SFCC-based monitoring at multiple Canadian sites, not in the fundamental physics of soil freezing.

**Reviewer #2's comment**

The authors repeatedly make the claim that there exists so called "traditional binary approaches" to describing soil frozen state, and "traditional monitoring methods, which rely on a binary 0 °C soil temperature threshold". I may not fully understand which approaches are referred to as "traditional". In my knowledge, now for decades in permafrost science and monitoring, we have not been limiting the description of ground state to frozen or unfrozen and in fact, great efforts have been directed to quantitatively and accurately describe soil partially frozen state. I think this

misunderstanding could be because the authors confound two distinct definitions used in permafrost context. [...]

**Response:**

We appreciate the reviewer's clarification. By "traditional binary methods," we intended to refer specifically to remote-sensing evaluation practices that classify soil as frozen or unfrozen based solely on temperature thresholds—typically 0 °C from land-surface model or in-situ soil temperature data. As correctly noted, in permafrost and seasonally frozen-ground studies, it has long been standard to distinguish between unfrozen, partially frozen, and fully frozen states using soil-specific freezing and thawing curves.

We have therefore revised the manuscript to remove all references to "traditional binary approaches" and to eliminate any suggestion that this three-state framework is new. As discussed in Comment 2, our intent was to emphasize that, unlike purely thermal classifications, the SFCC approach quantifies the *water-phase composition* of the soil and provides a continuous measure of freezing that can be directly derived from dielectric measurements.

To prevent confusion between the water/ice phase and thermal (cryotic/non-cryotic) definitions of frozen ground, the revised version ensures that our work focuses on the frozen state of the topsoil—where the state of water is the key variable—rather than on the thermal classification commonly used in permafrost mapping. The paragraph in the original *Introduction* that introduced the term "traditional binary" and compared these definitions has been entirely removed (the same paragraph that was discussed in Comment 2), as it was unnecessary and diverted attention from the main objective of our study.

**Changes made:**

- Deleted the paragraph in the *Introduction* that introduced the "traditional binary" terminology (see previous comment for modified text).
- Removed all remaining instances of the phrase "traditional binary approaches" throughout the manuscript.

• Clarified that the study concerns the monitoring of seasonally frozen ground through SFCCs, not the cryotic/non-cryotic thermal state relevant to permafrost classification.

**Reviewer #2's comment**

The authors seem to be unaware of evidence contradicting their assumptions about equal freezing and thawing curve patterns in in-situ measurements (Line 153–154). Field studies by Overduin et al. (2006) and Tomaškovičová & Ingeman-Nielsen (2024) showed very strong hysteresis effects in in-situ measurements of unfrozen soil moisture using dielectric permittivity sensors. It is possible that these effects may not be in fact related to real unfrozen water content difference between freezing vs. thawing branches of soil moisture curve at the same temperature (Wu, 2017), but nevertheless, the apparent hysteresis does affect in-situ soil moisture measurements based on electric principles.

**Response:**

We appreciate this important remark and acknowledge the significance of hysteresis between freezing and thawing curves. We are well aware of this phenomenon, as two of our co-authors have discussed it extensively in prior publications (Pardo Lara et al., 2020; Mavrovic et al., 2020).

In the initial submission, we assumed equivalence between the SFCC (Soil Freezing Characteristic Curve) and STCC (Soil Thawing Characteristic Curve) for the practical purpose of constructing a continuous annual time series of soil freezing probabilities. This assumption was informed by Pardo Lara et al. (2020), who reported weak hysteresis in the uppermost 5 cm of soil, in contrast to fine-grained, saturated permafrost samples where stronger hysteresis has been observed (e.g., Overduin et al., 2006; Tomaškovičová and Ingeman-Nielsen, 2024).

However, we agree that applying the SFCC to thawing periods introduces uncertainty—particularly after snowmelt—when additional water inputs violate the constant

total-water-content assumption required for curve construction. We have therefore revised the manuscript to limit the analysis exclusively to freezing cycles.

To address this comment, we implemented two major revisions:

• Focused exclusively on freezing cycles: We now explicitly state in the *Methodology* and *Discussion* sections that only the freezing periods are analyzed. We also clarify that reliable STCCs could not be constructed *in situ* due to hydrological inputs during thawing, especially from snowmelt, which invalidate the closed-system assumption of the SFCC model.

**Modified text in manuscript (Methodology – data preprocessing):**

.... Specifically, any fluctuations within  $\pm 2\sigma_T$  of 0°C, where  $\sigma_T$  represents the instrument-specific temperature uncertainty, were ignored (see Appendix ?? for details on sensor uncertainty). If, during a freezing cycle, the soil temperature never dropped below the  $-\sigma_T$  threshold and  $\varepsilon_{\rm eff}$  remained relatively unchanged, we classified these sites as never frozen. Although curve fitting was not feasible for these sites cycles due to insufficient data in Zones 2 and 3, they were retained for further analysis to investigate the freezing process across our monitoring networks. In this analysis, we subsequent analysis of freeze monitoring across our networks. We assumed that the total water content in the system remained equal to the initial water content and did not change during the freezing or thawing processes (He and Dyck, 2013). We monitored  $\varepsilon_{\rm eff}$  throughout both freezing and thawing cycles to validate this assumption. We interpreted significant, sudden surges in  $\varepsilon_{\mathrm{eff}}$  as indicators of additional water entering the system, violating this assumption. Consequently, we excluded such cycles from further analysis. While this assumption generally held during freezing cycles, it was often invalid during thawing cycles, primarily due to snowmelt introducing substantial amounts of water into the soil. As a result, the SFCC could be reliably constructed for freezing cycles, but constructing the STCC from in situ measurements during thawing cycles was often not feasible. Since this study aimed to use site- and cycle-specific SFCC/STCC to determine the degree of soil freezing and classify soil states, we applied the SFCC from each freezing cycle to the subsequent thawing cycle at the same site. Although differences between SFCC and STCC primarily driven by hysteresis effects—are pronounced in laboratory settings, previous studies (Pardo Lara et al., 2020; Mavrovic et al., 2020) found no visually discernible differences in situ. We assumed that the SFCC could reliably represent the STCC until a significant change in total soil water content, marked by a sudden surge in  $\varepsilon_{\text{eff}}$ , which approximately corresponds to the snowmelt event. To assess the reliability of applying the SFCC for thawing cycles, we evaluated its performance over the full thawing cycle and up to the snowmelt date. Therefore, in this study we focused exclusively on freezing cycles, excluding thawing cycles from further analysis.

**Modified text in manuscript (Discussion - exclusion of thawing period):**

The thawing cycles, however, were not analyzed in this study because constructing the STCC from in situ measurements is not reliably feasible. During thawing, snowmelt and rainfall introduce additional water into the soil, violating the constant total water content assumption required for curve development

• Added explicit literature review of hysteresis (Introduction): A new section was added to the *Introduction* acknowledging the occurrence of hysteresis

and referencing the key studies mentioned by the reviewer (Overduin et al., 2006; Tomaškovičová & Ingeman-Nielsen, 2024).

**Modified text in manuscript (Introduction – hysteresis):**

[...] especially under field conditions. The hysteresis between SFCC and STCC has been widely documented, particularly under laboratory conditions (Mavrovic et al., 2020; Pardo Lara et al., 2020; Wu et al., 2017)In situ studies, such as Tomaškovičová and Ingeman-Nielsen (2024), reported approximately 10% higher unfrozen water content during freezing than during thawing at equivalent temperatures under saturated, fine-grained permafrost conditions, reflecting latent heat effects, cryosuction, and slow pore-water redistribution. Similarly, Overduin et al. (2006) documented asymmetric freezing-thawing transitions in saturated silty clays caused by latent heat and moisture migration. By contrast, near-surface, non-permafrost measurements (Pardo Lara et al., 2020) found hysteresis to be theoretically expected but generally weak or indistinguishable.

Together, these revisions ensure that the revised manuscript explicitly addresses hysteresis effects, properly defines the scope of our analysis, and justifies the exclusion of thawing periods to maintain methodological consistency.

**Reviewer #2's comment**

Conclusion is made that the presented work will improve monitoring of permafrost, but it is unclear to me if this is for field or remote-sensing monitoring. The authors mention only field measurements, and they don't seem to make the link to the remote-sensing applications. For improving field measurements, again, the claim seems stretched, especially when insisting on the "binary classification" of soil moisture state.

**Response:**

We initially had some difficulty understanding this comment, as our study specifically addresses seasonally frozen ground, and the term permafrost is not mentioned anywhere in the manuscript. That said, we agree with the reviewer that certain statements in the original version may have been overstated. The study is primarily focused on *in-situ* monitoring of seasonally frozen ground across different Canadian ecozones using the SFCC framework. The revised version now ensures that our main focus remains on

field-based observations and analysis rather than on improving soil freezing monitoring in general.

We have also revised the *Conclusions* section to clearly state how the dataset generated from this study can support remote-sensing validation practices. Specifically, the dataset provides ground-truth information that includes transitional (partially frozen) states, which are often underrepresented in current evaluation datasets that rely mainly on air/soil temperature measurements. This clarification better links our in-situ work to its practical application in remote-sensing contexts.

**Changes made:**

- Removed overstated claims about "improving monitoring." The title was updated to emphasize the study's focus on in-situ monitoring.
- Clarified throughout that this work is a field-based study and not intended to improve soil freezing monitoring directly.
- Revised the *Conclusions* section to explicitly mention the dataset's relevance for remote-sensing evaluation, highlighting its inclusion of transitional soil states.

**Modified text in manuscript (Conclusions excerpt):**

This is particularly important given the rapid warming of high-latitude regions and the need for ground-truth evaluation of satellite-based freeze-thaw products, which currently rely primarily on air or soil temperature observations for training and evaluation (Rautiainen et al., 2025; Donahue et al., 2023; Roy et al., 2020; Kou et al., 2017; Gao et al., 2020

**Reviewer #2's comment**

Another conclusion claims to be able to quantify the degree of soil freezing. However, I do not see evidence of quantitative analysis in the work which appears limited to the qualitative description of soil as unfrozen, transitional and frozen.

**Response:**

We appreciate this observation and acknowledge that our initial manuscript did not sufficiently highlight the quantitative nature of our approach. In the revised version, we clarify that the freezing probability represents a continuous, quantitative measure of the degree of soil freezing. This probability is derived by propagating uncertainty from the hierarchical posterior distributions of  $T_f$  and b through the normalized SFCC formulation.

To make this explicit, the following revisions were implemented:

• Revised Methodology Section (Probability of Frozen Ground) to more clearly and concisely describe how  $P_{\text{frozen}}$  is computed and interpreted as a quantitative measure of soil freezing.

**Modified text in manuscript - methodology:**

The probability of frozen ground (hereafter referred to as the freezing probability), which can also be interpreted as the degree of soil freezing at the network level, was computed by propagating uncertainty from the hierarchical posterior distributions of  $T_f$  and b. For each network, paired posterior samples  $\{(T_f^{(s)}, b^{(s)})\}_{s=1}^S$  were drawn from the PyMC hierarchical models, where  $b^{(s)} = \exp(b_j^{(s)})$  restores the parameter to its original scale. Soil temperature observations were perturbed according to sensor uncertainty,  $T_{\text{sim}}^{(s)} \sim \mathcal{N}(T_{\text{obs}}, \sigma_T^2)$ , where  $\sigma_T$  was assigned based on the sensor type. For each posterior draw, the freezing probability was evaluated using the normalized SFCC, and the results were averaged across all Monte Carlo samples to obtain the mean freezing probability at each timestamp.

• Updated Figures 6–9 to display time series of freezing probability alongside soil and air temperature, visually emphasizing its continuous and quantitative nature.

Modified figure in manuscript (Results):

• Clarified the Results section to explain that the framework enables quantitative comparison of soil freezing behavior across networks and ecozones.

These clarifications ensure that readers can recognize  $P_{\text{frozen}}$  as a quantitative metric of soil freezing rather than a qualitative classification.

**Reviewer #2's comment**

Additionally, there appear to be a number of misconceptions, or perhaps poorly presented concepts, reiterated throughout the manuscript. For example, the concept of zero curtain (isothermal process of phase change between water and ice) appears to be confounded with the transitional zone, which encompasses a wider temperature (and liquid water content) range, based on Figures 6–10.

**Response:**

We agree with the reviewer's observation. In the revised manuscript, we ensured that the concepts of the zero curtain and the transitional zone are clearly distinguished. The

zero curtain refers to the period when soil temperature remains near 0 °C due to latent heat effects during the phase change between water and ice. In contrast, the transitional zone (Zone 2 of the SFCC) represents the broader temperature range over which ice and liquid water coexist, encompassing a wider range of liquid water contents.

To eliminate any ambiguity, the manuscript now focuses exclusively on the transitional state as defined within the SFCC framework. All mentions of the zero-curtain effect have been clarified or removed where they could be confused with the transitional zone.

**Changes made:**

- Removed or clarified all instances where the zero-curtain and transitional-zone concepts were conflated.
- Focused the discussion on the transitional state as defined by the SFCC framework.
- Ensured consistent and precise terminology throughout the manuscript.

**Reviewer #2's comment**

"Notwithstanding these remarks, I do think that the dataset seems extremely interesting, and worthy of publication, if adequately exploited."

**Response:**

We sincerely thank the reviewer for recognizing the value of the dataset. In the revised manuscript, we have taken several steps to better exploit and present its potential.

**Revisions implemented:**

- Applied Bayesian hierarchical partial pooling to derive more stable parameter estimates across sites and networks within ecozones.
- Revised the Conclusions section to clearly emphasize the dataset's potential applications in model development and remote-sensing validation.
- Committed to publishing the dataset publicly following manuscript acceptance to facilitate its reuse by the broader cold-regions and remote-sensing communities.

**References**

- Orlando Andersland and Branko Ladanyi. Frozen ground engineering. John Wiley & Sons, 2003.
- Kyle A. Arndt, Josh Hashemi, Susan M. Natali, Luke D. Schiferl, and Anna-Maria Virkkala. Recent Advances and Challenges in Monitoring and Modeling Non-Growing Season Carbon Dioxide Fluxes from the Arctic Boreal Zone. *Current Climate Change Reports*, 9(2):27–40, October 2023. ISSN 2198-6061. doi: 10.1007/s40641-023-00190-4. URL https://link.springer.com/10.1007/s40641-023-00190-4.
- George Bouyoucos and M. M. McCool. A New Method of Measuring the Concentration of the Soil Solution Around the Soil Particles. *Science*, 42(1084):507–508, October 1915. ISSN 0036-8075, 1095-9203. doi: 10.1126/science.42.1084.507. URL https://www.science.org/doi/10.1126/science.42.1084.507.
- Chris Derksen, Xiaolan Xu, R. Scott Dunbar, Andreas Colliander, Youngwook Kim, John S. Kimball, T. Andrew Black, Eugenie Euskirchen, Alexandre Langlois, Michael M. Loranty, Philip Marsh, Kimmo Rautiainen, Alexandre Roy, Alain Royer, and Jilmarie Stephens. Retrieving landscape freeze/thaw state from Soil Moisture Active Passive (SMAP) radar and radiometer measurements. Remote Sensing of Environment, 194: 48–62, June 2017. ISSN 00344257. doi: 10.1016/j.rse.2017.03.007. URL https://linkinghub.elsevier.com/retrieve/pii/S0034425717300998.
- Wojciech Dobiński. Permafrost active layer. Earth-Science Reviews, 208:103301, September 2020. ISSN 00128252. doi: 10.1016/j.earscirev.2020.103301. URL https://linkinghub.elsevier.com/retrieve/pii/S0012825220303470.
- Kellen Donahue, John S. Kimball, Jinyang Du, Fredrick Bunt, Andreas Colliander, Mahta Moghaddam, Jesse Johnson, Youngwook Kim, and Michael A. Rawlins. Deep learning estimation of northern hemisphere soil freeze-thaw dynamics using satellite multi-frequency microwave brightness temperature observations. *Frontiers in Big Data*, 6:1243559, November 2023. ISSN 2624-909X. doi: 10.3389/fdata.2023.1243559. URL https://www.frontiersin.org/articles/10.3389/fdata.2023.1243559/full.

- Huiran Gao, Ning Nie, Wanchang Zhang, and Hao Chen. Monitoring the spatial distribution and changes in permafrost with passive microwave remote sensing. *ISPRS Journal of Photogrammetry and Remote Sensing*, 170:142–155, December 2020. ISSN 09242716. doi: 10.1016/j.isprsjprs.2020.10.011. URL https://linkinghub.elsevier.com/retrieve/pii/S0924271620302823.
- Hailong He and Miles Dyck. Application of Multiphase Dielectric Mixing Models for Understanding the Effective Dielectric Permittivity of Frozen Soils. *Vadose Zone Journal*, 12(1):vzj2012.0060, 2013. ISSN 1539-1663. doi: 10.2136/vzj2012. 0060. URL https://onlinelibrary.wiley.com/doi/abs/10.2136/vzj2012.0060. \_eprint: https://onlinelibrary.wiley.com/doi/pdf/10.2136/vzj2012.0060.
- Youngwook Kim, John S. Kimball, Kyle C. McDonald, and Joseph Glassy. Developing a Global Data Record of Daily Landscape Freeze/Thaw Status Using Satellite Passive Microwave Remote Sensing. *IEEE Transactions on Geoscience and Remote Sensing*, 49(3):949–960, March 2011. ISSN 0196-2892, 1558-0644. doi: 10.1109/TGRS.2010. 2070515. URL http://ieeexplore.ieee.org/document/5599863/.
- Xiaokang Kou, Lingmei Jiang, Shuang Yan, Tianjie Zhao, Hui Lu, and Huizhen Cui. Detection of land surface freeze-thaw status on the Tibetan Plateau using passive microwave and thermal infrared remote sensing data. *Remote Sensing of Environment*, 199:291–301, September 2017. ISSN 00344257. doi: 10.1016/j.rse.2017.06.035. URL https://linkinghub.elsevier.com/retrieve/pii/S0034425717302948.
- Alexandre Roy. Soil dielectric characterization at L-band microwave frequencies during freeze-thaw transitions, July 2020. URL https://hess.copernicus.org/preprints/hess-2020-291/hess-2020-291.pdf.
- Alex Mavrovic, Oliver Sonnentag, Juha Lemmetyinen, Carolina Voigt, Nick Rutter, Paul Mann, Jean-Daniel Sylvain, and Alexandre Roy. Environmental controls of winter soil carbon dioxide fluxes in boreal and tundra environments. *Biogeosciences*, 20(24):5087–5108, December 2023. ISSN 1726-4170. doi: 10.5194/bg-20-5087-2023. URL https://bg.copernicus.org/articles/20/5087/2023/. Publisher: Copernicus GmbH.

- P.P. Overduin, D.L. Kane, and W.K.P. Van Loon. Measuring thermal conductivity in freezing and thawing soil using the soil temperature response to heating. *Cold Regions Science and Technology*, 45(1):8–22, June 2006. ISSN 0165232X. doi: 10.1016/j.coldregions.2005.12.003. URL https://linkinghub.elsevier.com/retrieve/pii/S0165232X06000048.
- R. Pardo Lara, A. A. Berg, J. Warland, and Erica Tetlock. In Situ Estimates of Freezing/Melting Point Depression in Agricultural Soils Using Permittivity and Temperature Measurements. *Water Resources Research*, 56(5): e2019WR026020, 2020. ISSN 1944-7973. doi: 10.1029/2019WR026020. URL https://onlinelibrary.wiley.com/doi/abs/10.1029/2019WR026020. \_eprint: https://onlinelibrary.wiley.com/doi/pdf/10.1029/2019WR026020.
- Kimmo Rautiainen, Manu Holmberg, Juval Cohen, Arnaud Mialon, Mike Schwank, Juha Lemmetyinen, Antonio De La Fuente, and Yann Kerr. An operational SMOS soil freeze—thaw product. *Earth System Science Data*, 17(10):5337–5353, October 2025. ISSN 1866-3516. doi: 10.5194/essd-17-5337-2025. URL https://essd.copernicus.org/articles/17/5337/2025/.
- Alexandre Roy, Peter Toose, Alex Mavrovic, Christoforos Pappas, Alain Royer, Chris Derksen, Aaron Berg, Tracy Rowlandson, Mariam El-Amine, Alan Barr, Andrew Black, Alexandre Langlois, and Oliver Sonnentag. L-Band response to freeze/thaw in a boreal forest stand from ground- and tower-based radiometer observations. *Remote Sensing of Environment*, 237:111542, February 2020. ISSN 00344257. doi: 10.1016/j.rse.2019.111542. URL https://linkinghub.elsevier.com/retrieve/pii/S0034425719305620.
- Joshua P. Schimel and Carl Mikan. Changing microbial substrate use in Arctic tundra soils through a freeze-thaw cycle. *Soil Biology and Biochemistry*, 37(8):1411–1418, August 2005. ISSN 00380717. doi: 10.1016/j.soilbio.2004.12.011. URL https://linkinghub.elsevier.com/retrieve/pii/S0038071705000209.
- Shahabeddin Taghipourjavi, Christophe Kinnard, and Alexandre Roy. Sentinel-1-Based Soil Freeze—Thaw Detection in Agro-Forested Areas: A Case Study in Southern

- Québec, Canada. Remote Sensing, 16(7):1294, April 2024. ISSN 2072-4292. doi: 10.3390/rs16071294. URL https://www.mdpi.com/2072-4292/16/7/1294.
- Huihui Tian, Changfu Wei, Houzhen Wei, and Jiazuo Zhou. Freezing and thawing characteristics of frozen soils: Bound water content and hysteresis phenomenon. *Cold Regions Science and Technology*, 103:74–81, July 2014. ISSN 0165232X. doi: 10.1016/j. coldregions.2014.03.007. URL https://linkinghub.elsevier.com/retrieve/pii/S0165232X14000627.
- Soňa Tomaškovičová and Thomas Ingeman-Nielsen. Quantification of freeze—thaw hysteresis of unfrozen water content and electrical resistivity from time lapse measurements in the active layer and permafrost. *Permafrost and Periglacial Processes*, 35 (2):79–97, April 2024. ISSN 1045-6740, 1099-1530. doi: 10.1002/ppp.2201. URL https://onlinelibrary.wiley.com/doi/10.1002/ppp.2201.
- Yuxin Wu, Seiji Nakagawa, Timothy J. Kneafsey, Baptiste Dafflon, and Susan Hubbard. Electrical and seismic response of saline permafrost soil during freeze Thaw transition. Journal of Applied Geophysics, 146:16–26, November 2017. ISSN 09269851. doi: 10.1016/j.jappgeo.2017.08.008. URL https://linkinghub.elsevier.com/retrieve/pii/S0926985117302914.
- Mingyi Zhang, Xiyin Zhang, Jianguo Lu, Wansheng Pei, and Chong Wang. Analysis of volumetric unfrozen water contents in freezing soils. *Experimental Heat Transfer*, 32 (5):426–438, September 2019. ISSN 0891-6152, 1521-0480. doi: 10.1080/08916152.2018. 1535528. URL https://www.tandfonline.com/doi/full/10.1080/08916152.2018. 1535528.
- T. Zhang and R. L. Armstrong. Soil freeze/thaw cycles over snow-free land detected by passive microwave remote sensing. *Geophysical Research Letters*, 28(5):763–766, March 2001. ISSN 0094-8276, 1944-8007. doi: 10.1029/2000GL011952. URL https://agupubs.onlinelibrary.wiley.com/doi/10.1029/2000GL011952.